

# A multiphysical ensemble system of numerical snow modelling.

Matthieu Lafaysse, Bertrand Cluzet, Marie Dumont, Yves Lejeune, Vincent Vionnet, and Samuel Morin

Météo-France - CNRS, CNRM UMR3589, Centre d'Études de la Neige (CEN), Grenoble, France

*Correspondence to:* matthieu.lafaysse@meteo.fr

**Abstract.** Physically based multilayer snowpack models suffer from various modelling errors. To represent these errors, we built the new multi-physical ensemble system ESCROC (Ensemble System Crocus) by implementing new representations of different physical processes in the deterministic coupled multi-layer ground/snowpack model SURFEX/ISBA/Crocus. This ensemble was driven and evaluated at Col de Porte (1325 m a.s.l., French alps) over 18 years with a high quality meteorological and snow dataset. 7776 simulations were evaluated separately accounting for the uncertainties of evaluation data. The ability of the ensemble to capture the uncertainty associated to modelling errors is assessed for snow depth, snow water equivalent, bulk density, albedo and surface temperature. Different sub-ensembles of the ESCROC system were studied with probabilistic tools to compare their performance. Results show that optimal members of the ESCROC system are able to explain more than half of the total simulation errors. Integrating members with biases exceeding the range corresponding to observations uncertainties is necessary to obtain an optimal dispersion, but this issue can also be a consequence of the fact that meterorological forcing uncertainties were not accounted for. ESCROC is a promising system to integrate numerical snow modelling errors in ensemble forecasting and ensemble assimilation systems in support of avalanche hazard forecasting and other snowpack modelling applications.

## 1 Introduction

Operational forecasting of avalanche hazard requires an analysis of current snowpack state and meteorological forecasts for the upcoming days, in order to estimate future snow conditions. Several organizations in different countries have implemented snowpack modelling approaches for this purpose (Durand et al., 1999; Lehning et al., 1999; Vikhamar-Schuler et al., 2011; Bellaire and Jamieson, 2013). The goal is to provide a description of the space and time variability of the current snowpack which is out of reach using observations only and to anticipate its evolution according to the expected meteorological conditions a few hours to days ahead. Meteo-France has been operating a deterministic snowpack forecasting system for about 20 years. The current version is based on a chain of 3 models: SAFRAN-SURFEX/ISBA/Crocus-MEPRA (S2M, Durand et al., 1999; Lafaysse et al., 2013). It simulates the snowpack evolution over all the French massifs for a large range of elevations, slope values and aspects. The meteorological analysis and forecasting system SAFRAN (Durand et al., 1993) provides meteorological data at a 1000 km$^2$ scale for a given elevation since the beginning of the season with a 2 day forecast lead time. These data are used to force the SURFEX/ISBA/Crocus snowpack model (Vionnet et al., 2012) which simulates the detailed snowpack stratigraphy for the different elevations and slopes. The mechanical stability of each snow profile is assessed by the





MEPRA expert system (Giraud, 1992). This system is supposed to help the forecasters in their decision process. However, the large discrepancies between this simulation system and the observed or perceived conditions in the field limit its pratical interest relatively to empirical considerations based on snow stratigraphy and surface properties measurements and the outputs of Numerical Weather Prediction (NWP) models. Similar issues are met by the other organizations operating systems based

either on SURFEX/ISBA/Crocus or on the SNOWPACK snow model of similar complexity (Bartelt and Lehning, 2002).

    Snowpack modelling systems suffer from various types of uncertainties. First, the estimated meteorological conditions since the beginning of the season are uncertain because meteorological analysis systems such as SAFRAN only assimilate scattered meteorological observations and suffer from the errors of their guess, usually analyses or forecasts of NWP models. NWP models errors, due to both initial conditions approximations and simplifications in atmospheric modelling are also responsible

for significant errors of the meteorological forcing for the forecast period. Then, uncertainties are associated with the spatial resolution of the systems, much coarser than the spatial scale of the snowpack variability involved in avalanche release mechanisms. Finally, snowpack modelling inherently contains simplifications and parameters uncertainties.

    It is necessary to better quantify the uncertainties resulting from these different errors for two main reasons. First, an objective quantification of uncertainties helps forecasters assess the confidence they may have in a specific model forecast on a specific

date. Second, a better knowledge of uncertainties is necessary to reduce model error by using assimilation data techniques to correct the simulated snowpack state with ground-based or remotely-sensed snow observations. In other disciplines such as meteorology and hydrology, ensemble approaches are now generalized (Swinbank et al., 2016), especially for the furthest prediction lead times when the uncertainty is increasing. Ensemble simulations are also the basis for the confidence in future climate simulations (IPCC, 2013). Several assimilation data techniques are also based on building ensembles to characterize

guess errors. Therefore, ensemble approaches are a natural candidate to quantify uncertainties in numerical snow modelling. Vernay et al. (2015) illustrated that the uncertainty of the meteorological forecast can be taken into account by forcing a snowpack model by ensemble NWP systems and that it improves the robustness of the snowpack simulations compared to the deterministic approach. Charrois et al. (2016) implemented an ensemble assimilation algorithm able to correct snowpack simulations using remotely-sensed visible spectral reflectances of the surface if the ensemble captures all the possible states of

the snowpack. To generalize these methods, all the uncertainties involved in numerical snow modelling must be accounted for in ensemble systems. Accounting for all the main uncertainties (meteorological forecast, initial conditions, model structure) has been shown to improve the overall skill of ensemble simulations for hydrology (Thiboult et al., 2016).

    Concerning the uncertainty attached to the snowpack model, Essery et al. (2013) proposed the first attempt to build an ensemble of 1701 snow simulations based on different combinations of physical options for various processes in a common

model structure called JIM (JULES Investigation Model). A reduced version of 32 members (FSM for Factorial Snow Model) was also published by Essery (2015). This system was designed to investigate the relative contribution of each main physical process in the spread usually obtained in model intercomparison projects. Therefore, the JIM ensemble spread describes the uncertainty resulting from currently available multi-layer snow models of different complexities. The authors identified minimal requirements for a satisfactory snowpack model (e.g. prognostic representation of snow albedo and density), but they also

conclude that there is not a single best member for all criteria and seasons. Their results also demonstrate that evaluations



restricted to snow depth measurements only and/or to a small number of seasons can lead to misleading conclusions. Raleigh et al. (2015) compared the spread of the JIM ensemble with ensemble simulations resulting from disturbed meteorological scenarios and found that the uncertainty in meteorological data prevails in general applications of numerical snow modelling except in very well instrumented observatories where snow model structure becomes the main source of uncertainty. The JIM

ensemble spread was not characterized from a probabilistic point of view (adequate dispersion for ensemble applications, i.e. representative of model errors) because it was not designed for that. Note also that the JIM ensemble is not appropriate for avalanche hazard forecasting because it does not include a detailed description of snow stratigraphy and snow metamorphism, and because some of the members include options which neglect processes known to be critical for mechanical stability such as compaction or liquid water storage.

The goal of this paper is to investigate the uncertainty involved in the different processes of a detailed multilayer snowpack model (SURFEX/ISBA/Crocus) in order to explore the possibility of building an ensemble version of this model suited for snowpack ensemble forecast and snowpack ensemble assimilation applications. Our aim differs from that of Essery et al. (2013) because we expect this ensemble to consist of members with the same overall degree of complexity and type of output, and to include a detailed stratigraphy of the snowpack allowing applications such as avalanche hazard forecasting, among

others. Ideally the skill of one given member should not be significantly poorer than another one in a general perspective (for various variables, sites, seasons and periods of the year). The spread of this ensemble should be of the same magnitude as that of typical model errors (Fortin et al., 2015) on different evaluation variables. This should allow correct characterization of the uncertainty in real time applications. Such a system would also allow accounting for the snowpack model error in snow hydrology and in future snowpack projections driven by future climate scenarios.

The major difficulty in building such an ensemble is that it requires isolating the uncertainty originating from the snowpack model itself from the other uncertainty sources. Therefore, following the conclusions of Raleigh et al. (2015) and similarly to Essery et al. (2013), our strategy is to build and evaluate our ensemble on one specific well instrumented site where we assume that input meteorological errors are low compared to model errors. It also allows to follow Essery et al. (2013) recommendations in terms of evaluation data because numerous snow variables are available over a long period. Sect. 2 summarizes the dataset

used to force and evaluate a new multi-physical version of the Crocus snowpack model called ESCROC (Ensemble System CROCus) and presented in Sect. 3. The evaluation methodology (Sect. 4) includes both a classical deterministic evaluation member by member and a probabilistic evaluation of the ensemble skill. The results are presented in Sect. 5 for different sub-ensembles. The limitations of the "perfect meteorological forcing" assumption are discussed in Sect. 6, as well as the impacts of the other limitations of our methodology on the possible applications of ensemble numerical modelling of snow.

## 2 Dataset

### 2.1 Forcing data

The 18-year long snow and meteorological dataset from the Col de Porte observatory (CDP, 1325 m altitude, 45.3ºN, 5.77ºS) is used to force and evaluate ESCROC from 1993 to 2011. A complete description of the dataset is provided by Morin et al.



(2012). The CDP site is a grassy meadow characterized by its mid-altitude and located in the Chartreuse massif in the French Alps. Snow is usually present from December to April, but snow melt and rainfall events can occur any time in the snow season. Wind is usually low, so that drifting events are unusual. The quality and completeness of this dataset have allowed to drive and evaluate various snowpack models (e.g. Magnusson et al., 2015; Decharme et al., 2016, a complete list is available

at http://www.umr-cnrm.fr/spip.php?article533). All the forcing variables of the Crocus model are directly measured: incident shortwave and longwave radiations, 2 m air temperature and specific humidity, 10 m wind speed and precipitation rates. Precipitation phase is manually assessed using all possible ancillary information. The only modification applied to the original published dataset is a bias correction of the incident longwave correction. Indeed, a break was identified in the time series in December 2010, associated with a replacement of the sensor. A -10 $\mathrm{Wm^{-2}}$ homogeneous correction is applied before December

2010 and a +10 $\mathrm{Wm^{-2}}$ homogeneous correction is applied from December 2010 onwards. These values were estimated from the concomitant evolution of the monthly mean difference between these data and the longwave incident radiation simulated by the Ritter and Geleyn (1992) radiative transfer scheme included in SAFRAN meteorological reanalyses (Durand et al., 2009). Although this correction is lower than the manufacturer uncertainty of 10% (Morin et al., 2012), it has a significant impact on snow simulations (not shown), consistent with the results of Raleigh et al. (2015) and Sauter and Obleitner (2015). The other

errors of the forcing data, described in Morin et al. (2012), are neglected in this study as previously mentioned.

## 2.2    Evaluation data

Six variables were selected to evaluate ESCROC: snow depth (SD), snow water equivalent (SWE), bulk density (BD), surface broadband albedo (A), snow surface temperature (SST) and ground temperature at 50 cm depth (GT50). To eliminate the summer period without snow on the ground, the time series are limited to the period between October 1[st] and June 30[th]. The

0 values of SD and SWE between these two dates are kept for the evaluations to appropriately evaluate the formation and disappearance of the snowpack. BD, A, and SST are not defined when there is no snow on the ground. For the 6 variables, Fig. 1 shows histograms of differences between distinct observations of the same variable in the experimental plot, which are used to quantify the uncertainty of each observed variable. Data observed at CDP since 2011 were sometimes used for observation error analysis but not for model forcing and evaluation.

### 2.2.1    Snow depth (SD)

Snow depth data from ultra-sound depth gauge span the whole winter from 1993 to 2010, and was completed by laser ranger data for the 2010-2011 winter. The instrumental error of these sensors (about 1 cm) is low compared to the spatial variability of snow depth in the meadow. The latter was estimated comparing ultra-sound sensor data with weekly snowpit measurements at 3 different locations (Fig. 1a). The differences usually range between 0 and 20 cm.





**Figure 1.** Distribution of differences between the reference observations and other sources of data: (a) Snow depth (SD) automatic measurements vs the three snowpits (2001-2011); (b) Snow water equivalent (SWE) automatic measurements vs the snowpit SWE measurements that are not used for sensor calibration (2001-2011); (c) Bulk density (BD) from the reference snowpit vs the other snowpits (2001-2011); (d) daily reference albedo vs data from Dumont et al. (2016) (winter 2013-2014); (e) reference snow surface temperature (SST) vs SST from the hemispheric pyrgeometer (1993-2010); (f) Reference ground temperature (GT) at -20 cm vs measurements of a new sensor at the same depth (season 2013-2014).





### 2.2.2 Snow water Equivalent (SWE)

Snow water equivalent is measured on a daily basis by cosmic rays sensors (NRC) that are linearly calibrated every year to match weekly snowpit measurements (Gottardi et al., 2013). This time series is available since 2001. The sensor accuracy (about 10%) is of the same magnitude as the spatial variability in the meadow that can be derived from snow pits data (Fig. 1b). A similar magnitude of uncertainty was found by Smith et al. (2016).

### 2.2.3 Bulk density (BD)

Although both SD and SWE automatic measurements are available daily since 2001, their spatial variability in the meadow and therefore between the two sensor locations is too high to compute a robust daily time series of snow density, especially with shallow snow cover and during the melting season. For this reason, bulk density was computed from weekly snowpit measurements of SWE and SD. Fig. 1c illustrates the uncertainty in this data resulting from its spatial variability (3 snow pits on each date). The instrumental error of the snow core sampling is not well known. Proksch et al. (2016) give a magnitude of 1 to 5% for average density measurements errors with smaller cutters and a small number of data. Conger and McClung (2009) obtained an uncertainty range of 2-11% for individual layer density measurements and emphasized that their accuracy depends on the cutter size. In the present study, the instrumental error of large cutters for bulk density is estimated to be roughly 10 kg m$^{-3}$.

### 2.2.4 Snow Albedo (A)

Daily broadband albedo was calculated from daily averaged downwelling and upwelling broadband shortwave radiation fluxes and is discarded during snowfall events, or when the measured fluxes are too low (Morin et al., 2012). Albedo data under 0.6 were also discarded in this work as it is an evidence of patchy snow cover. The resulting data are rather discontinous, however spans the entire 1993-2011 period. It is difficult to quantify the uncertainty of these data directly from the pyranometer manufacturer values. The uncertainty was estimated by comparing the data with broadband albedo data obtained from spectral measurements during the winter 2013-2014 (Dumont et al., 2016). The differences (Fig. 1d) originate from both the sensor errors and the spatial variability in the plot. They are quite high relatively to the range of possible values for snow.

### 2.2.5 Snow Surface Temperature (SST)

Two values for the hourly mean snow surface temperature were derived separately from the upcoming longwave radiation and from an infrared sensor with a narrower angle of view, both assuming an emissivity of 1 for snow as described in Morin et al. (2012). An additional data treatment was performed to discard artefacts due to patchy snow covers or to a heating of the metal mast: temperatures exceeding the melting point of water, some outliers under 240 K and times where A < 0.6 or SD < 0.3m were removed from the time series. As this variable exhibits a high diurnal cycle, we decided to keep 12:00 UTC and 24:00 UTC temperature measurements in order to balance between daytime and nighttime values. The availability of two distinct



long time series allows to quantify the uncertainty in SST measurements resulting from both instrumental errors and spatial variability in the meadow (Fig. 1e).

### 2.2.6 Ground temperature at 50 cm (GT50)

Ground temperature is recorded at 3 different depths (-10, -20 and -50 cm). Without snow, the diurnal cycle is very high close to the surface but it is not a key process to simulate later the heat flux at the interface with the snowpack, more affected by long-term biases. Therefore, we preferred to select the 50 cm depth for the evaluations. Only the October-December period was used for this variable as it is the key period for snowpack simulations. Spring and summer errors are much less important and ground temperature stays usually very close to 0°C between January and the end of the season. No other sensor provides temperature measurement at this depth. New sensors were however installed in October 2012 at few meters away at 5, 10 and 20 cm depth. We assume that the differences between the sensors at 20 cm (Fig. 1f) provide an estimate of the magnitude of the uncertainty resulting from spatial variability and instrumental accuracy. The distance between sensors is probably too low but the spatial variability is expected to decrease with depth.

## 3 The ESCROC multiphysical snow model

The SURFEX/ISBA/Crocus unidimensional multi-layer physical snowpack model is extensively described in Vionnet et al. (2012). It computes all terms of the energy and mass balance of the snowpack. In the standard version, the prognostic variables of each snow layer are mass, density, temperature, liquid water content, semi-empirical variables describing the snow microstructure, and the layer age (as a proxy for the amount of impurities). The lagrangian discretization rules are designed to accurately reproduce the snowpack stratigraphy evolution, a key component for avalanche forecasting. The heat flux at the ground-snow interface is simulated by a semi-implicit coupling with the multi-layer ground model ISBA-DIF (Decharme et al., 2011).

In NWP ensemble modelling, two options are usually employed to disturb the model physics: stochastic perturbations of parameters (Palmer et al., 2009) or multiple combinations of several physical options, commonly called mutliphysics (Charron et al., 2010; Descamps et al., 2014). In ESCROC, the multiphysics approach has been chosen. The ESCROC multiphysical ensemble is built implementing new physical options inside the SURFEX/ISBA/Crocus source code for the main physical processes, resulting in a "factorial" ensemble as in Essery et al. (2013). Several options implemented in the source code in previous studies (Carmagnola et al., 2014; Charrois et al., 2016) but not used in the standard version are incorporated in the ensemble. New options extracted from the literature and existing snowpack models such as SNOWPACK or JIM were also implemented for other physical processes. Modifications of especially uncertain and sensitive parameters are also introduced for processes whose representations available in the literature seem to not cover sufficiently the uncertainty. The selected options for each process are summarized in Fig. 2 and detailed below.





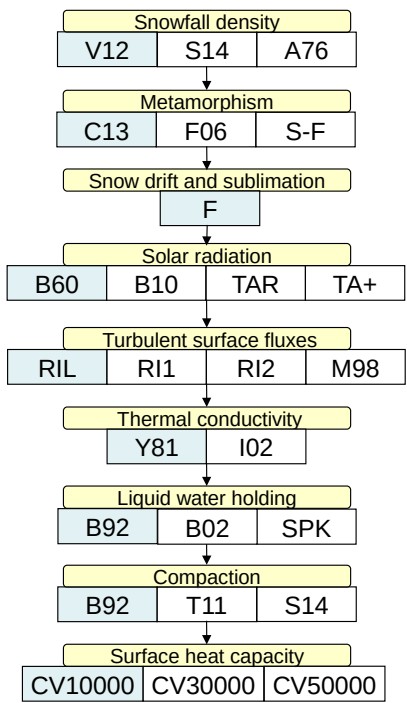

**Figure 2.** ESCROC physical options, blue cells correspond to the default Crocus configuration.

## 3.1 Fresh snow properties

Crocus computes density and microstructural properties of falling snow from air temperature and wind speed during snowfall. Different laws are available in the literature for fresh snow density. All of them are empirical, relying on simultaneous measurements of wind speed and air temperature followed by manual measurements of freshly fallen snow density. Crocus default option called V12 (Vionnet et al., 2012; Pahaut, 1975) computes density $\rho_n$ from 2 m air temperature $T_a$ (in °C) and 10 m wind speed $U$ (m s$^{-1}$):

$$\text{V12:} \quad \rho_n = \max(\rho_{\min}, a_\rho + b_\rho T_a + c_\rho U) \tag{1}$$

The parametrization of density from SNOWPACK (Schmucki et al., 2014), here called S14 was also implemented. It depends also on wind speed and air temperature:

$$\text{S14:} \quad \begin{cases} \log_{10}(\rho_n) = e_\rho + f_\rho T_a + g_\rho + h_\rho \sin^{-1}(\sqrt{i_\rho}) + j_\rho \log_{10}(\max[U,2]) & \text{if } T \geq -14^o C \\ \log_{10}(\rho_n) = e_\rho + f_\rho T_a + h_\rho \sin^{-1}(\sqrt{i_\rho}) + j_\rho \log_{10}(\max[U,2]) & \text{otherwise} \end{cases} \tag{2}$$





**Table 1.** Parameter values of fresh snow density options

| Option | Parameters |
|---|---|
| V12 | $\rho_{\min} = 50 \text{ kg m}^{-3}$ ; $a_\rho = 109 \text{ kg m}^{-3}$ ; $b_\rho = 6 \text{ kg m}^{-3}\text{K}^{-1}$ ; |
|  | $c_\rho = 26 \text{ kg m}^{-7/2}\text{s}^{-1/2}$ ; $d_\rho = 8 \text{ kg m}^{-3}\text{K}^{-1}$ |
| S14 | $e_\rho = 3.28$ ; $f_\rho = 0.03 \text{ K}^{-1}$ ; $g_\rho = -0.36$; $h_\rho = -0.75$ ; $i_\rho = 0.8$ ; $j_\rho = 0.3$ |
| A76 | $\rho_{\min} = 50 \text{ kg m}^{-3}$ ; $k_\rho = 1.7 \text{ K}^{-1}$ ; $l_\rho = 15 \text{ K}$ |

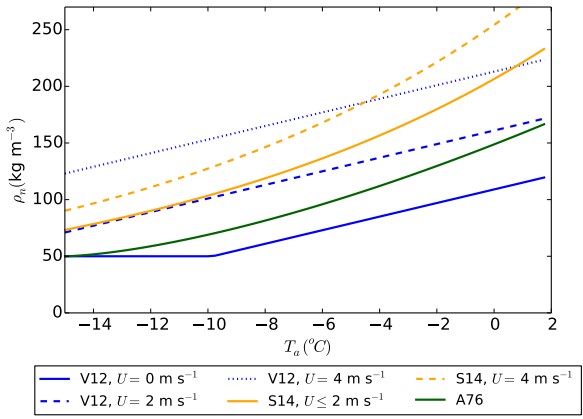

**Figure 3.** Fresh snow density as a function of air temperature and wind for the 3 options included in ESCROC.

Finally, we implemented the law from Anderson (1976) where the density only depends on air temperature:

$$\text{A76:} \quad \rho_n = \rho_{\min} + \max(k_\rho(T_a + l_\rho)^{1.5}, 0) \tag{3}$$

The parameters of the three laws are given in Table 1 and their behaviour is summarized in Fig. 3.

The uncertainty associated with the initial microstructural properties is not included in ESCROC because the sensitivity of
5   albedo (and in turn of the energy balance) to the snow specific surface area (SSA) is low for typical values of fresh snow SSA
usually above $40 \text{ m}^2\text{kg}^{-1}$ (Gallet et al., 2009).

### 3.2 Snow metamorphism

The default Crocus representation of snow metamorphism (B92) describes the snow microstructure by a semi-quantitative
formalism with dendricity, sphericity and size (Brun et al., 1992). The time evolution of these continous parameters follows
10   empirical laws whose parameters were adjusted through experimental investigations. In the case of dry snow, these laws depend
mostly on temperature and temperature gradient. For wet snow metamorphism, evolution laws only depend on liquid water
content (Brun et al., 1989). An alternative formalism was introduced by Carmagnola et al. (2014) who replaced dendricity



and size by the optical diameter, which is measurable in the field. However, as the formalism is a translation of B92 laws with new prognostic variables, using both formulations is almost equivalent and not able to describe the uncertainty associated with this process. Therefore, only the C13 option is included in ESCROC. We also included the time evolution law of the optical diameter from Flanner and Zender (2006) which fits the model outputs of a snow microstructure model representing

the diffusive vapour fluxes among the grains (F06). This law was also implemented by Carmagnola et al. (2014) who found that its skill was similar to the C13 formulation. Note that this option does not affect wet metamorphism, which still follows the original formulation of Brun et al. (1989).

We also implemented the formulation of Schleef et al. (2014) who focused on the 48 first hours after snowfall and proposed the following law experimentally calibrated under isothermal conditions:

$$\frac{dSSA}{dt} = (a + bT)SSA^m \qquad (4)$$

with $a = 1.1E - 6$, $b = 3.1E - 8$ and $m = 3.1$

As this formulation only applies to the first 48 hours, we chose to associate this option with the F06 evolution law for older snow (option S-F).

### 3.3   Blowing snow and associated sublimation

Wind-induced snow transport drift is a complex phenomenon known to be a key factor for slab avalanches as it creates local snow accumulation and has an effect on metamorphism reducing the grain size, compacting the top layers and increasing their cohesion. Sublimation of snow particles in turbulent suspension is frequent during blowing snow events. A scheme to represent erosion and accumulation on the different aspects of idealized slopes can be activated in Crocus (Durand et al., 2001). For more general applications where erosion and accumulation cannot be simulated by a unidimensional model, Vionnet et al. (2012)

introduced a parameterization to modify surface layers snow properties during blowing snow events and to optionally simulate mass loss due to blowing snow drift. Although these options can be very sensitive in windy environments (Libois et al., 2015), we decided to deactivate it for the first version of ESCROC because it is not possible to evaluate the uncertainty of this process at CDP where blowing snow is almost never observed.

### 3.4   Solar radiation absorption

The original formulation of radiative transfer in the snowpack (Brun et al., 1992) consists in computing solar radiation absorption and reflexion in three spectral bands (0.3-0.8; 0.8-1.5 and 1.5-2.8 $\mu$m) and is partly inspired from the work of Warren (1982). Only the grains characteristics of the uppermost 2 numerical layers are used to compute the albedo of each spectral band. A parameterization reduces the albedo in the visible band as a function of the age of the layer and the altitude of the site to mimic the effect of light-absorbing impurities. This parameterization relies on a time constant $\tau_a$ usually taken as 60 days at

mid-latitudes (option B60). At CDP, this value is suspected of partially explaining a positive bias in Crocus simulated albedo. Therefore, we define the B10 option by using a 10-day value.





**Table 2.** Parameter values for solar radiation absorption options

| Option | Parameters |
|---:|---|
| B60 | $\tau_a = 60$ days |
| B10 | $\tau_a = 10$ days |
| TAR | $c_0 = 5$ ng g$^{-1}$ ; $\tau_0 = 4$ ng g$^{-1}$ day$^{-1}$ ; $l_f = 0.05$ m |
| TA+ | $c_0 = 50$ ng g$^{-1}$; $\tau_0 = 200$ ng g$^{-1}$ day$^{-1}$ ; $l_f = 0.05$ m |

The radiative transfer scheme TARTES (Two-streAm Radiative TransfEr in Snow, Libois, 2014) has recently been implemented in Crocus (and used by Libois et al. (2015) and Charrois et al. (2016)). It simulates the absorption of solar radiation within the stratified snowpack using the $\delta$-Eddington approximation and with a spectral resolution of 20 nm. This scheme explicitly accounts for the spectral and angular characteristics of the incident radiance (e.g. the solar zenith angle and the presence of a cloud cover) and uses an explicit impurity content for each layer (equivalent black carbon content). The impurity content $c$ is initialized to a constant concentration $c_0$ during snowfall. A dry deposition term $\tau_{dd}$ is added to the layer at depth D at each time step:

$$\tau_{dd} = \tau_0 e^{-D/l_f} \tag{5}$$

This formulation and its parameters are rather uncertain as it has not been specifically evaluated against observations. The typical magnitude of the parameters for black carbon only would be of $c_0 = 5$ ng g$^{-1}$ and $\tau_0 = 4$ ng g$^{-1}$ according to Doherty et al. (2013) or Sterle et al. (2013) (option TAR), but values 20 times as high were sometimes observed. Furthermore, as other impurities such as dust and vegetal debris are not explicitly accounted for, higher values of the parameters are possible to indirectly simulate the effect of these species with their black-carbon equivalent concentrations. We included in ESCROC two different sets of parameters for this physical scheme (TAR and TA+). The 4 resulting options for solar radiation absorption are summarized in Table 2.

### 3.5 Turbulent Fluxes

As detailed in Vionnet et al. (2012), Crocus surface turbulent fluxes (latent heat flux $LE$ and sensible heat flux $H$ ) are proportional to the turbulent exchange coefficient or drag coefficient $C_H$. $C_H$ depends on the Richardson number $Ri$, which quantifies the stability of the atmosphere directly above the snow surface, and on the effective roughness length for momentum $z_0$ and for heat $z_{0h}$ (Louis, 1979; Noilhan and Mahfouf, 1996). In particular $C_H$ decreases when $Ri$ increases (Fig. 4), and the rougher the surface, the higher the fluxes. The Louis parameterization tends to minimize surface heat fluxes under stable conditions ($Ri > 0$) but in mountainous terrain with complex topography, mechanical or orographic-induced turbulence is




**Table 3.** Parameter values for turbulent surface fluxes options

| Option | Parameters |
|---|---|
| RIL | $Ri_l = 0.2$ ; $z_0 = 10^{-3}m$ ; $z_{0h} = 10^{-4}m$ |
| RI1 | $Ri_l = 0.1$ ; $z_0 = 10^{-3}m$ ; $z_{0h} = 10^{-4}m$ |
| RI2 | $Ri_l = 0.026$ ; $z_0 = 10^{-3}m$ ; $z_{0h} = 10^{-4}m$ |
| M98 | $Ri_l = 0.026$ ; $z_0 = 10^{-3}m$ ; $z_{0h} = 10^{-3}m$ |

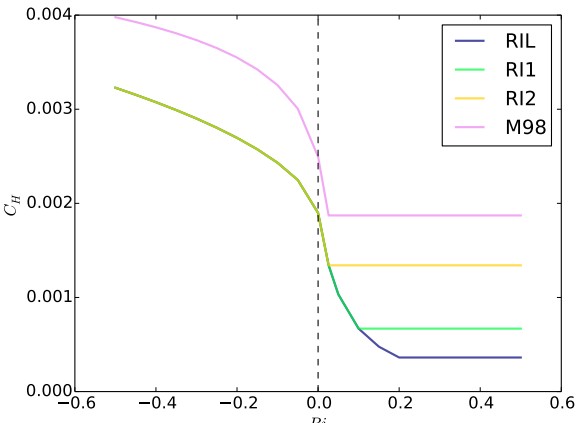

**Figure 4.** $C_H$ as a function of Ri for the 4 turbulent fluxes options included in ESCROC (RIL, RI1, RI2, M98) as summarized in table 3.

suspected of producing exchange rates higher than in this theory. For this reason, Martin and Lejeune (1998) proposed to apply in Crocus a threshold $Ri_l = 0.026$ on the Richardson number to maintain a minimal level of turbulence in stable conditions:

$$C_H(Ri \geq Ri_l) = C_H(Ri_l) \tag{6}$$

They also applied an effective roughness length for heat supposed to be closer to the roughness length for momentum than usual values for snow (option M98).

In the current default version of the model (RIL), Vionnet et al. (2012) chose the formulation of the ISBA-ES snow scheme (Boone and Etchevers, 2001), already included in the SURFEX platform, where $Ri_l$ is taken as 0.2 and $z_{0h}$ as $10^{-4}$ m. As preliminary studies showed that both configurations simulate turbulent fluxes of different magnitude, we address the uncertainty in ESCROC using also two intermediate configurations (RI1 and RI2) as summarized in Table 3 and Fig. 4.





**Table 4.** Parameter values for thermal conductivity options

| Option | Parameters |
|---|---|
| Y81 | $a_\lambda = 2.22$ W m$^{-1}$K$^{-1}$ ; $\lambda_{\min} = 4 \times 10^{-2}$ Wm$^{-1}$K$^{-1}$ |
| I02 | $e_\lambda = 2.0 \times 10^{-2}$ W m$^{-1}$K$^{-1}$ ; $f_\lambda = 2.5 \times 10^{-6}$ W m$^5$K$^{-1}$kg$^{-2}$ |
| | $g_\lambda = -6.023 \times 10^{-2}$ W m$^{-1}$K$^{-1}$ ; $h_\lambda = -2.5425$ W m$^{-1}$ |
| | $i_\lambda = -289.99$ K ; $P^0 = 10^5$ Pa |

## 3.6 Thermal conductivity

Snow thermal conductivity $\lambda$ mostly depends on density (Calonne et al., 2011). Two different options were already implemented in Crocus. The default option follows Yen (1981) (Y81):

$$\text{Y81:} \quad \lambda = \max \left[ a_\lambda (\frac{\rho_{snow}}{\rho_w})^{1.88} ; \lambda_{\min} \right] \tag{7}$$

This formulation is very close to the experimental law proposed by Calonne et al. (2011) based on tomographic images of snow (Fig. 5).

The other implemented option, I02, is the default formulation of the ISBA-ES snow model (Boone, 2002; Sun et al., 1999):

$$\text{I02:} \quad \lambda = e_\lambda + f_\lambda \rho_{snow}^2 + \left( g_\lambda + \frac{h_\lambda}{T + i_\lambda} \right) \frac{P^0}{P} \tag{8}$$

Parameter values are given in Table 4. I02 law depends not only on density but also on snow temperature and it has a higher

conductivity than experimental values (Fig. 5) to indirectly compensate for the fact that latent heat fluxes due to vapor fluxes are not represented in Crocus. This is expected to increase vertical heat transfer as temperature increases.

## 3.7 Liquid water content

Liquid water content is an important variable for its role on metamorphism, thermal exchanges due to phase changes, compaction, and mechanical stability. However, it is especially challenging to both observe and simulate liquid water percolation

in the snowpack particularly because its horizontal variability is very high at macroscopic scale. Work is in progress to include in Crocus a formulation solving Richards equations in snow as in Wever et al. (2014). However, the only formulation currently available for the liquid water percolation in the snowpack follows a simple and conceptual bucket approach where the layers are seen as superposed water reservoirs with a homogeneous volumetric liquid water content $w_{liq}$ in kg m$^{-3}$. When $w_{liq}$ exceeds the maximum liquid water holding capacity $w_{liq\,\max}$, the excess water drains to the underlying layer.

In the default version of Crocus (B92 option), the volumetric liquid water holding capacity is defined by a fixed maximal percentage of the pores volume (Pahaut, 1975):

$$\text{B92:} \quad w_{liq\,\max} = 0.05 \rho_w \phi \tag{9}$$




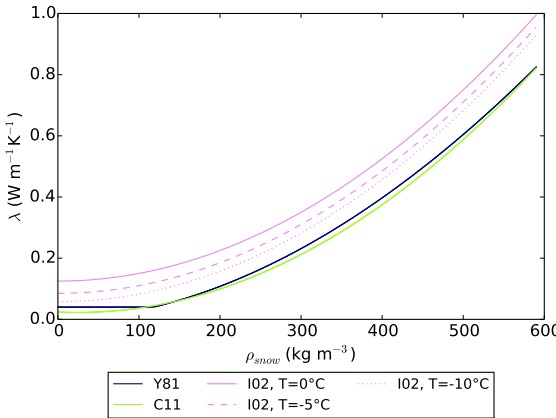

**Figure 5.** Snow thermal conductivity as a function of density for the 2 options included in ESCROC (Y81, I02), and for the experimental law proposed by Calonne et al. (2011) (C11).

.

where $\phi = 1 - \dfrac{\rho - w_{liq}}{\rho_i}$ is the snow porosity

$$(10)$$

Equations 9 and 10 should replace the formulas with typos given in Vionnet et al. (2012) and in Gascon et al. (2014). As a result, the higher the density the lower the maximal volumetric liquid water content.

The experiments of Coléou and Lesaffre (1998) give water holding capacities about 40% higher than B92 formulation as shown in Fig. 6. There was also a typo in the equation of that paper, corrected in equation 11.

C98: $\quad w_{liq\,\max} = \rho * \left( 0.057 \dfrac{\phi}{1-\phi} + 0.017 \right)$

$$(11)$$

Based on this finding, a similar formulation was chosen for the bucket version of the SNOWPACK model. Equation 12 replaces equation 1 of Wever et al. (2014) where there is a typo in the conditions:

W14: $\quad \begin{cases} w_{liq\,\max} = \rho_w * (0.08 - 0.1023(0.97 - \phi)) & \text{if } \phi \geq 0.77 \\ w_{liq\,\max} = \rho_w * \left( 0.0264 + 0.0099 \frac{\phi}{1-\phi} \right) & \text{otherwise} \end{cases}$

$$(12)$$

This option was also included in ESCROC (W14).

In several other models (Essery et al., 2013), the water retention capacity is defined by a maximal liquid water mass fraction. We included in ESCROC the ISBA-ES formulation (B02, Boone, 2002) for which this threshold is set to $r_{\min} = 0.03$ for snow densities above $\rho_r = 200$ kg m$^{-3}$ and to higher values up to 0.05 for very low density following:

B02: $\quad w_{liq\,\max} = \dfrac{\rho}{\rho_w} \left( r_{\min} + (r_{\max} - r_{\min}) \max \left( 0, \dfrac{\rho_r - \rho}{\rho_r} \right) \right)$

$$(13)$$





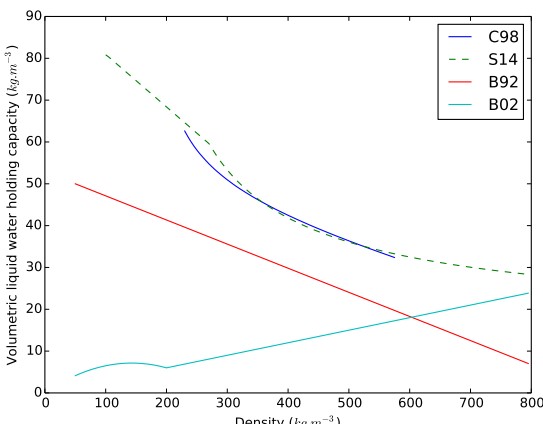

**Figure 6.** Volumetric liquid water holding capacity as a function of the layer density for the 3 options included in ESCROC (B92, S14, B02) and for the experimental law proposed by Coléou and Lesaffre (1998) (C98).

where $r_{\max} = 0.1$

This formulation has an opposite behaviour: the higher the density, the higher the maximal volumetric liquid water content (Fig. 6). The discrepancies between B02 and B92/W14 options increase for low densities. Although such a behaviour is not consistent with laboratory experiments of Coléou and Lesaffre (1998), including this formulation in ESCROC allows to very indirectly account for the uncertainty linked to the heterogeneous percolation of liquid water which result in the presence of wet layers below layers that are not homogeneously saturated, increasing the percolation velocity. Note that liquid water percolation is also affected by capillary effects and by a high dependence of snow permeability on the pores structure as reported by Jordan et al. (1999) or Calonne et al. (2012). As these processes cannot be described by the bucket approach used in Crocus, it is reasonable to include in ESCROC a wide uncertainty for low densities by the use of these 3 different formulations.

## 3.8 Compaction

For a given layer of thickness D the settling under the over burden $\sigma$ is expressed as (Anderson, 1976; Navarre, 1975) :

$$\frac{\mathrm{d}D}{D} = \frac{-\sigma}{\eta}\mathrm{d}t \qquad (14)$$

The viscosity $\eta$ is expressed as a function of density $\rho$ and temperature $T$ (°$C$) by:

$$\text{B92:} \quad \eta = f_1(w_{liq})f_2(g_s)\eta_0\frac{\rho}{c_\eta}e^{a_\eta(-T)+b_\eta\rho} \qquad (15)$$

where $\eta_0 = 7.6223710^6$ kg s$^{-1}$m$^{-1}$, $a_\eta = 0.1$ K$^{-1}$, $b_\eta = 0.023$ m$^3$ kg$^{-1}$ and $c_\eta = 250$ kg m$^{-3}$. Two multiplicative functions $f_1$ and $f_2$ allow representing respectively the faster settlement of wet snow as a function of the volumetric liquid water content




$w_{liq}$, and the reduced settlement of layers consisting of faceted snow types especially depth hoar as a function of grain size $g_s$ (Vionnet et al., 2012).

However, the temperature and density dependence of snow viscosity in the general case have various forms in the literature as reviewed by Teufelsbauer (2011) who proposed a new law fitting the data of separate experimental works. If this law is combined with Crocus parameterization for wet snow and depth hoar, we can formulate a new option T11 as:

$$\text{T11:} \quad \eta = f_1(w_{liq})f_2(g_s) * 0.05\rho^{-0.0371T+4.4}(10^{-4}e^{0.018\rho} + 1) \tag{16}$$

The compaction velocity however has a complex dependence to snow microstructure (Lehning et al., 2002) which cannot be described by the representation of snow microstructure in Crocus. We account for this source of uncertainty by implementing the S14 option where we apply a non-linear relationship between settlement, the stress $\sigma$ $(Pa)$ and the SSA decrease from Schleef et al. (2014) for the 48 first hours after snowfall:

$$\text{S14:} \quad \frac{\mathrm{d}\rho}{\mathrm{d}t} = B\rho\frac{\mathrm{d}SSA}{\mathrm{d}t}\sigma^k \tag{17}$$

with $B = -6.6 \times 10^{-3}$ and $k = 0.18$. The current Crocus parameterization is applied when the snow layer age exceeds 2 days.

### 3.9 Soil scheme

The conductive heat flux at the soil-snow interface depends on the temperature gradient between the snow bottom and the upper soil layer. It is explicitly modeled by a semi-implicit soil-snow coupling with one of the ISBA soil scheme options. Although several options are available in the SURFEX platform (Masson et al., 2013), the multi-layer diffusive approach [ISBA-DIF] (Decharme et al., 2011, 2013) is preferred to force-restore options (Noilhan and Mahfouf, 1996) which are not able to represent the seasonal heat storage in the deep soil and therefore a realistic heat flux at the soil-snow interface (Habets et al., 2003). In the general case, the uncertainties in soil modelling come from both the representation of physical processes and the soil texture and vegetation parameters. Their comprehensive exploration would represent a full extended new work beyond the scope of this paper. Sensitivy tests suggested that one of the most sensitive parameter on snow simulations is the surface heat capacity $c_T$ (J kg$^{-1}$ K$^{-1}$) used in the soil surface temperature $T_S$ prognostic equation:

$$\frac{\partial T_S}{\partial t} = \frac{1}{c_T}G - \frac{1}{c_{G_1}}\frac{\lambda_1}{\Delta z_1}(T_S - T_2) \tag{18}$$

where $G$ is the sum of radiative and turbulent energy fluxes at the surface (Wm$^{-2}$), $\lambda_1$ the thermal conductivity of the first soil layer (W m$^{-1}$K$^{-1}$), $\Delta z_1$ its depth (m), $c_{G_1}$ its heat capacity, and $T_2$ the temperature of the second soil layer. For the CDP grassy meadow, the surface heat capacity is equal to the vegetation heat capacity $c_V$. The default value of $c_V$ has been set to $5\times10^4$ J kg$^{-1}$ K$^{-1}$ since the 1990s for numerical stability reasons in the coupling with NWP models but this is unrealistic when compared with values from the literature (Dupont et al., 2014). In the v8.0 SURFEX release (http://www.umr-cnrm.fr/surfex/ /spip.php?rubrique148), the current default value has been set to $10^4$ J kg$^{-1}$ K$^{-1}$ for low lying vegetation. Although more realistic, at Col de Porte this value gives a significant cold bias in autumn, regardless of the snow scheme used, as illustrated




by Fig. 5 of Decharme et al. (2016). In ESCROC, we account for the uncertainty in the soil-snow heat flux by including 3 different values of the vegetation heat capacity: $10^4$, $3 \times 10^4$ and $5 \times 10^4$ J kg$^{-1}$ K$^{-1}$. However, we must keep in mind that this unrealistic range probably compensates other model errors.

## 4 Evaluation methodology

### 4.1 Members scores

Although all ESCROC options are supposed to represent the corresponding physical processes with an equivalent complexity and reliability, it is necessary to check if each combination of options has a satisfactory overall skill or if some of them should be eliminated. The skill of each singular member is evaluated by deterministic scores comparing $N$ simulated values $m_{k_i}$ by member $i$ to the corresponding $N$ observations $o_k$. Thus, we compute the bias estimator $\widehat{B_i}$ and the Root Mean Square Error estimator $\widehat{\mathrm{RMSE}}_i$:

$$\widehat{B_i} = \frac{1}{N} \sum_{k=1}^{N} (m_{k_i} - o_k) \tag{19}$$

$$\widehat{\mathrm{RMSE}}_i = \sqrt{\frac{1}{N} \sum_{k=1}^{N} (m_{k_i} - o_k)^2} \tag{20}$$

These scores are called estimators because they give an uncertain description of the true model skill for several reasons. First, the reference observations $o_k$ represent an uncertain description of the site-scale snow conditions due to both instrumental errors and spatial variability at the site. Then, the skill of a member with respect to observations has a significant variability from one season to another so that uncertainty is associated with the limited available evaluation period. To account for these uncertainties, we model a member score $S_i$ as a random variable following a Normal distribution $\mathcal{N}(\widehat{S_i}, \widehat{\sigma_i})$, where the total variance is decomposed by $\widehat{\sigma_i^2} = \sigma_o^2 + \sigma_{p_i}^2$. To quantify the $\sigma_{p_i}$ component corresponding to the evaluation period uncertainty, we applied bootstrapping (Efron, 1979) over annual time series on each variable: for each member, instead of computing only one score on the available years (1993-1994,1994-1995,1995-1996,...,2010-2011), 1000 different score estimators were computed from 1000 18-year long bootstrap samples obtained by draw with replacement of the available years such as (1998-1999,1994-1995,1994-1995,...,2003-2004). $\sigma_{p_i}$ is the standard deviation of these estimators. In the general case, the observation uncertainty $\sigma_o$ is difficult to quantify without a large range of sensors measuring instrumental uncertainty and spatial uncertainty and it is often necessary to choose a priori values. In this study, for SD, SWE, albedo, SST and GT the mean and the standard deviation of the differences between the reference observation and other data from another instrument located at another place in the plot (Sect. 2.2, Fig. 1) allow to give an estimate of $\sigma_o$ for the bias and the RMSE. For BD, the estimated instrumental error has to be added to the spatial uncertainty shown in Fig. 1c. The values of $\sigma_o$ are summarized in Table 5.



**Table 5.** Values of $\sigma_o$ for the bias and the RMSE resulting from Sect. 2.2

| Variable | $\sigma_o$ for bias | $\sigma_o$ for RMSE |
|---|---|---|
| SD (cm) | 9.0 | 12.2 |
| SWE (kg m$^{-2}$) | 16 | 38 |
| BD (kg m$^{-3}$) | 10 | 26.5 |
| Albedo | 0.040 | 0.069 |
| SST (K) | 1.07 | 1.44 |
| GT50 (K) | 0.42 | 0.50 |

With the assumption of a Gaussian distribution of the score, it is possible to compute its 90% confidence interval and to test the significance of the difference between two different members by a Student test. Note that this Gaussian approximation is not realistic for RMSE but the high uncertainty of $\sigma_o$ estimates do not justify to propose a more sophisticated error modelling.

This methodology is applied for each evaluated variable listed in Sect. 2.2. For SD, it is also applied separately for early snow
depth (October-January) and late snow depth (from the date of observed maximum SWE to the end of the season) because the absence of bias in any part of the snow season is also required to consider a member as satisfactory.

### 4.2   Ensemble scores

The skill of the ensemble can be assessed both by deterministic scores applied to the ensemble mean and by probalistic scores describing the spread and the ability of the ensemble to compute reliable probabilities. The RMSE of an ensemble of population
$n$ corresponds to the RMSE of the ensemble mean $\bar{E}$. $\bar{E}$ is defined for each date k as the mean of the $p_k \leq n$ available members $(m_{k_i})_{i \in [1, p_k]}$ (Eq. 21). This restriction is necessary when computing scores on model variables that are not defined when there is no snow on the ground (BD, SST, A). For example, at the end of the snow season, when not all members have melted yet, the BD ensemble mean will be considered as the BD mean of the members that still have snow.

$$\bar{E}_k = \frac{1}{p_k} \sum_{i=1}^{p_k} m_{k_i} \tag{21}$$

The ensemble spread or dispersion $\sigma_E$ is the standard deviation of members, relatively to the ensemble average:

$$\sigma_E = \sqrt{\frac{1}{N} \sum_{k=1}^{N} \frac{1}{p_k} \sum_{i=1}^{p_k} (m_{k_i} - \bar{E}_k)^2} \tag{22}$$

The spread-skill SS of the ensemble $E$ of mean $\bar{E}$ is then defined as :

$$\mathrm{SS} = \frac{\sigma_E}{\mathrm{RMSE}(\bar{E})} \tag{23}$$

The ensemble dispersion is optimal if $SS = 1$ (Fortin et al., 2015). In that case, the spread of the ensemble is representative of
the error of the mean, and for an unbiased system, observations will be most of the time inside the ensemble envelope.




The Continous Ranked Probability Score (CRPS) is one of the most common probabilistic tool to evaluate the ensemble skill both in terms of reliability (unbiased probabilities) and resolution (ability to separate the probability classes) (Candille and Talagrand, 2005):

$$\mathrm{CRPS} = \frac{1}{N} \sum_{k=1}^{N} \int_{\mathbb{R}} (F_k(x) - H(x - o_k))^2 \mathrm{d}x \tag{24}$$

where $F_k(x)$ is the cumulative distribution function of the ensemble simulation at time $k$ and $H(y)$ is the Heaviside function ($H(y) = 0$ if $y \leq 0$; $H(y) = 1$ if $y > 0$). CRPS value has the same unit as the evaluted variable and tends towards 0 for a perfect system. It is mainly useful to compare the overall skill of several ensembles. A skill score (CRPSS) can be defined to compare the CRPS of an ensemble $E$ to a reference $R$:

$$\mathrm{CRPSS}(E) = 1 - \frac{\mathrm{CRPS}(E)}{\mathrm{CRPS}(R)} \tag{25}$$

To compare the skill of a multiphysics ensemble simulation to the more classical deterministic snow modelling, we choose a single member for the reference $R$. In such a case, the CRPS reduces to the Mean Absolute Error of the deterministic simulation. As the standard Crocus configuration (blue options in Fig. 2) is not optimal at Col de Porte, we choose the member with CV30000 surface heat capacity option and default other options as reference. This configuration has a better overall skill.

We also used rank histograms (Hamill, 2001) which illustrate the occurrence frequency of the different possible ranks of the observations $o_k$ among the sorted ensemble members. The flatness of this histogram is a condition of the system reliability (if the simulated probabilities are unbiased regardless of the probability level, the different ranks should have a uniform occurrence frequency). It is also an indicator of the spread-skill as underdispersion will result in a U-shaped rank histogram and overdispersion in a bell-shaped rank histogram.

Note that SS, CRPS, and rank histograms are also affected by observations uncertainties similarly to deterministic scores. As a first step, this uncertainty is not modelled here but this lack should be kept in mind in the comparison of different ensembles.

### 4.3 Members selection methods

As equiprobability of members is prefered in ensemble forecasting and ensemble assimilation applications, we extracted a sub-ensemble of $n$ equiprobable members, i.e. without a significantly lower skill than other members on any of the 8 evaluation variables. We first selected this sub-ensemble $E_1$ based on the deterministic evaluation of all ESCROC members described in Sect. 4.1, and evaluted its ensemble skill as described in Sect. 4.2 for 5 evaluation variables (SD, SWE, BD, A, SST). GT50 is not included for this evaluation step because ESCROC does not include a comprehensive description of uncertainty in soil modelling.

Then, among these equiprobable members, it is useful to limit the number of members $n'$ to keep reasonable computing-times in larger scales applications, and it may be possible to improve the spread-skill by selecting the most appropriate members since many configurations are highly correlated. We consider here that the value of $n'$ is prescribed as a technical constraint.




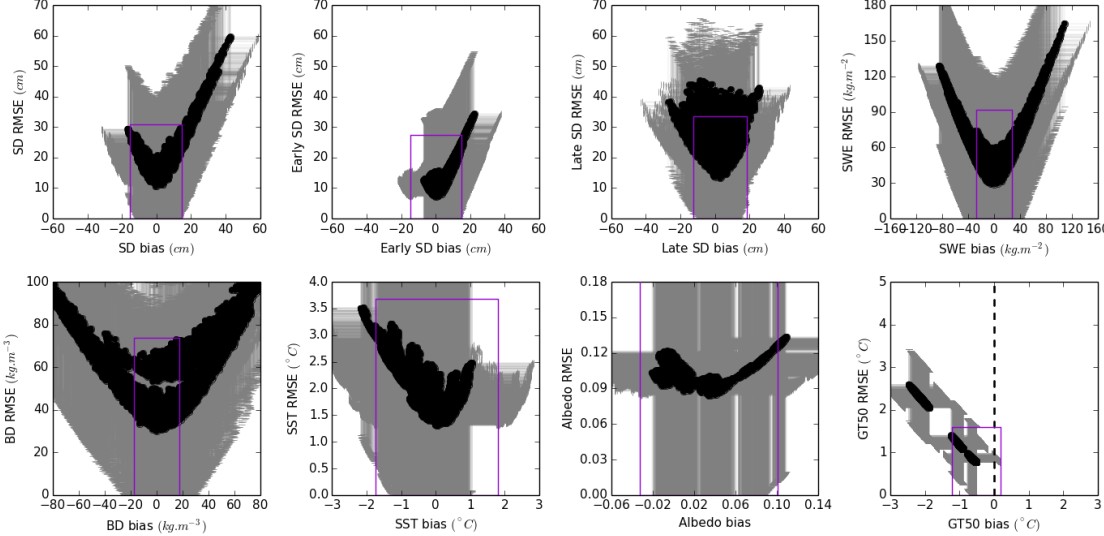

**Figure 7.** Scores of each member of the ESCROC ensemble for the different evaluation variables, 1993-2011. Black points represent the couple $(\widehat{B_i}, \widehat{\mathrm{RMSE}_i})$ for each member. Gray crosses represent the 90% confidence interval of each score. The violet rectangle represents the 90% confidence interval associated to the best member according to its RMSE.

For illustration, we take $n' = 35$ as in Vernay et al. (2015). Many possible criteria can be defined to select these $n'$ members. Here, we chose to present two selections based respectively on the optimization of the spread-skill for snow depth (ensemble $E_2$) and of the CRPS for snow depth (ensemble $E_3$). Other criteria might be better suited for specific applications and it would also be possible to define a multivariate criteria. In any case, the number of possible combinations $C_n^{n'} = \frac{n!}{n'!(n-n')!}$ can be

huge when $n >> n'$ and testing all the possible ensembles can become unrealistic. In such a case, we propose to test the optimization criteria (SS or CRPS here) randomly among 10000 combinations, and to evalute the obtained sub-ensembles $E_2$ and $E_3$. The solution will obviously be neither absolute nor unique. If the dispersion of the $E_1$ full ensemble of equiprobable members is not sufficient to obtain a satisfactory sub-ensemble $E_2$ in terms of dispersion, it is possible to include members with an individual lower skill among the initial candidates, but in such a case all members would not be equiprobable anymore.

We define ensemble $E_4$ by applying the selection method based on snow depth SS optimization in the specific case where all the 7776 ESCROC members are candidates.

## 5   Results

### 5.1   Deterministic evaluation of the members and definition of the sub-ensemble of equiprobable members $E_1$

Fig. 7 shows the couple of scores $(\widehat{B_i}, \widehat{\mathrm{RMSE}_i})$ of each ESCROC member and their confidence interval for the different

evaluation variables. For each evaluation variable, the members cover a continous range of bias and RMSE values. For snow





surface properties (albedo, SST), this range is lower than the width of the confidence intervals of the scores of the best members. Very few members can be excluded from the ensemble on this criterion. However, for integrated variables (early, late, and full-season snow depth, snow water equivalent, bulk density) and for the ground temperature, the range of the scores is larger than the confidence intervals of the best members. A significant number of members (30 to 80% depending on the variable) exhibit

scores included in the confidence interval (i.e. the skills of these members are not significantly different from a statistical point of view). Only 575 members (about 7%) exhibit, for all variables simultaneously, scores not significantly different from the best member. These members constitute the sub-ensemble $E_1$ as defined in Sect. 4.3.

## 5.2   Probabilistic evaluation of the different sub-ensembles

The sub-ensemble $E_1$ of $N_1 = 575$ optimal and equiprobable members is evaluated in terms of spread-skill and CRPS for the

different evaluation variables. For all the variables, the magnitude of the spread ranges between 40 and 60% of the RMSE of the ensemble mean (Table 6, first column). This means that about half of the total uncertainty is unexplained by this sub-ensemble. As a result, the observation is frequently not included in the ensemble as illustrated by the over-representation of the extreme ranks in rank diagrams for all variables (Fig. 8, first column), especially for albedo consistently with the lower SS of this variable. This behaviour is also illustrated in the temporal plots: in Fig. 9, fisrt column, the observed time series of snow

depth is more than 10% of the time outside the 90% interval confidence of the ensemble (occurrences above the ensemble in 2003-2004 and 2010-2011 and conversely below the ensemble in 2006-2007 and 2007-2008). Several occurrences outside the ensemble can also be seen for bulk density and albedo during the season 2007-2008 (Fig. 10). Despite this underdispersion, the positive CRPSS (Table 7) demonstrate that sub-ensemble $E_1$ exhibits a better skill than the deterministic approach from a probabilistic point of view. The interest of the ensemble framework is also illustrated in Fig. 9 and 10 where observations

are usually not superposed with the ensemble median but are found inside the ensemble spread with varying positions from one date to another. However, it is important to consider that the uncertainty in the evaluation data is not taken into account in this ensemble evaluation although it can impact the results. Thus, the unbalance between the first and last ranks frequencies in snow depth and SWE rank diagrams might suggest a positive bias of snow depth in $E_1$, but it is inconsistent with the negative bias of SWE and no significant bias of BD. This behaviour can only be explained by the uncertainty in the reference data.

Among these 575 optimal members, the sub-ensembles $E_2$ and $E_3$ of $N_2 = 35$ members optimized in terms of spread-skill or CRPS of snow depth are selected following the methodology of Sect. 4.3. The members are listed in Tables A1 and A2. The improved spread-skill of $E_2$, and even of $E_3$ (Table 6) illustrate that a much lower number of members is sufficient to represent model uncertainty if they are appropriately selected, and that including too correlated members tends to reduce the dispersion. However, despite the SS optimization, the sub-ensemble $E_2$ remains underdispersive for all variables. This is also the case for

ensemble $E_3$, a fortiori. The U-shape of rank diagrams is only slightly smoothed (Fig. 8) and the simulated time series feature the same main issues between $E_1$, $E_2$ and $E_3$ (Fig. 9 and 10). It can also be noticed that the optimization of either snow depth SS or snow depth CRPS degrades neither SS nor the CRPS for the other evaluation variables (Tab 6 and 7). The only exception is the CRPS of bulk density which is poorer for ensemble $E_3$ than for ensemble $E_1$.




**Table 6.** Spread-skill (Dispersion / RMSE) of ESCROC sub-ensembles for the different evaluation variables, 1993-2011.

| Variable | $E_1$ | $E_2$ | $E_3$ | $E_4$ |
|---|---|---|---|---|
| SD (cm) | 51% (8.5 / 16.5) | 65% (10.0 / 15.3) | 56% (8.0 / 14.3) | 100% (14.8 / 14.8) |
| SWE (kg m$^{-2}$) | 51% (38.6 / 75.2) | 69% (48.6 / 70.2) | 57% (38.1 / 66.9) | 131% (82.9 / 63.2) |
| BD (kg m$^{-3}$) | 60% (21.5 / 35.8) | 62% (22.0 / 35.5) | 62% (22.3 / 36.0) | 132% (45.7 / 34.7) |
| Albedo | 42% (0.034 / 0.081) | 46% (0.035 / 0.076) | 46% (0.035 / 0.076) | 52% (0.039 / 0.075) |
| SST (K) | 59% (1.03 / 1.74) | 61% (1.08 / 1.78) | 63% (0.98 / 1.54) | 70% (1.05 / 1.50) |

**Table 7.** CRPS (CRPSS in parenthesis) of ESCROC sub-ensembles for the different evaluation variables, 1993-2011.

| Variable | $E_1$ | $E_2$ | $E_3$ | $E_4$ |
|---|---|---|---|---|
| SD (cm) | 7.2 (0.41) | 6.6 (0.46) | 6.3 (0.48) | 6.1 (0.49) |
| SWE (kg m$^{-2}$) | 40.2 (0.30) | 36.8 (0.36) | 36.9 (0.36) | 32.8 (0.43) |
| BD (kg m$^{-3}$) | 19.4 (0.26) | 19.5 (0.25) | 22.3 (0.15) | 18.5 (0.29) |
| Albedo | 0.056 (0.09) | 0.052 (0.16) | 0.052 (0.16) | 0.050 (0.19) |
| SST (K) | 0.89 (0.47) | 0.92 (0.45) | 0.78 (0.53) | 0.76 (0.55) |

To increase the dispersion, it is possible to relax the constraint of optimal skill in terms of bias and RMSE for some (or all) evaluation variables for the candidate members. Ensemble $E_4$ corresponds to the extreme case: we selected 35 members within all the 7776 ESCROC members by snow depth SS optimization (list of members in Table A3). We can see that the spread-skill can be fully optimized for a chosen variable (perfect SS of snow depth in Table 6, much more flatter rank diagram in Fig. 8, and

5   significant broadening of the Q5-Q95 interquantile interval in Fig. 9 allowing the inclusion of the observed time series most of the time). However, it does not allow to simultaneously optimize the dispersion of the other variables: albedo and SST are still underdispersive whereas SWE and BD become overdispersive (see also the bell-shaped rank diagrams). It can be noticed that although members with a lower individual skill are included, the CRPS are still enhanced for all variables compared to other sub-ensembles even without any weighting of the less probable members. However, the significance of CRPS differences

10   among the different sub-ensembles was not tested here.

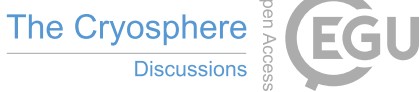

**Figure 8.** Rank diagrams for the different sub-ensembles ($E_1$, $E_2$, $E_3$, $E_4$, by column) and evalution variables (SD, SWE, BD, A, SST, by line), 1993-2011.





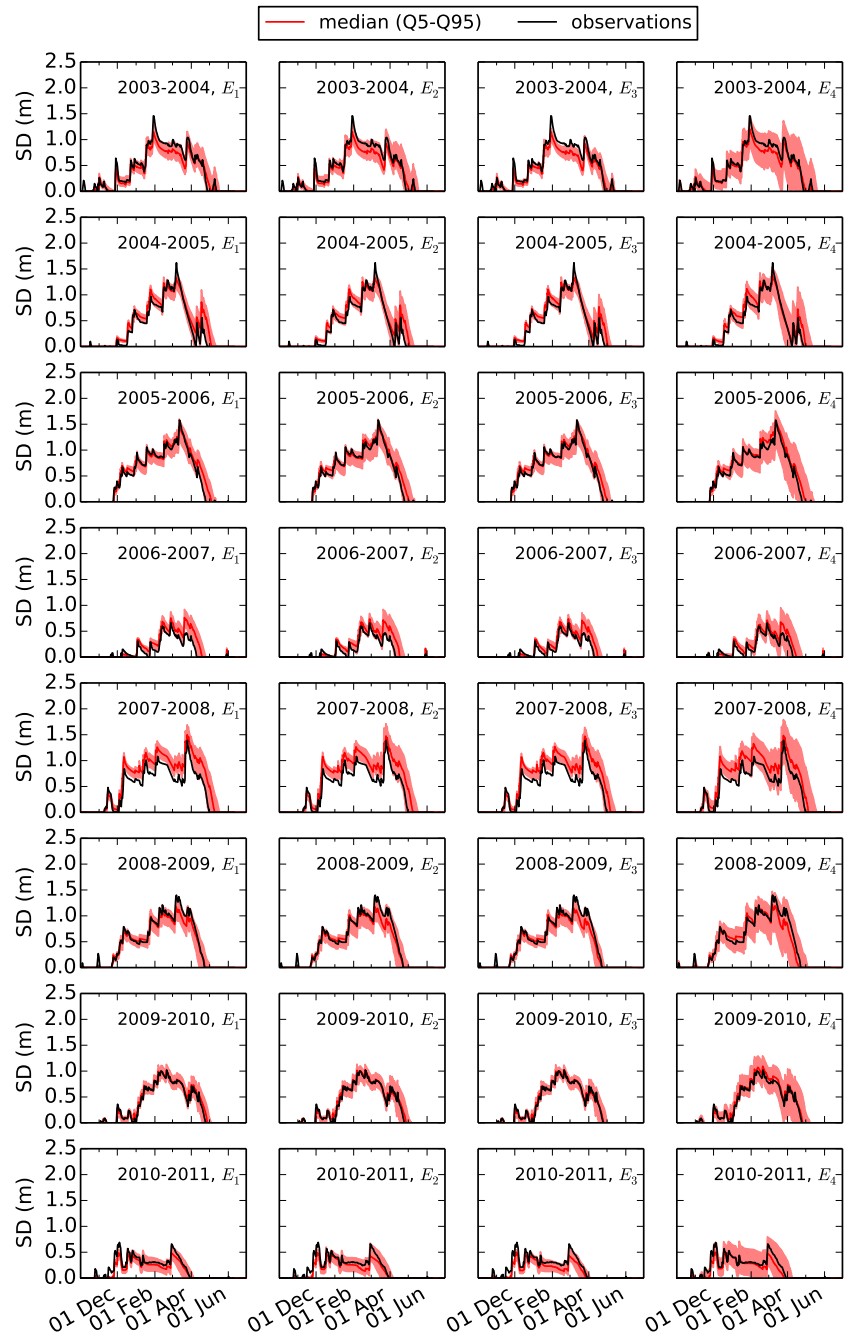

**Figure 9.** Simulated snow depth time series (median, 5th and 95th percentiles) for the different sub-ensembles ($E_1$, $E_2$, $E_3$, $E_4$, by column), 2003-2011. Black line: observed snow depth.





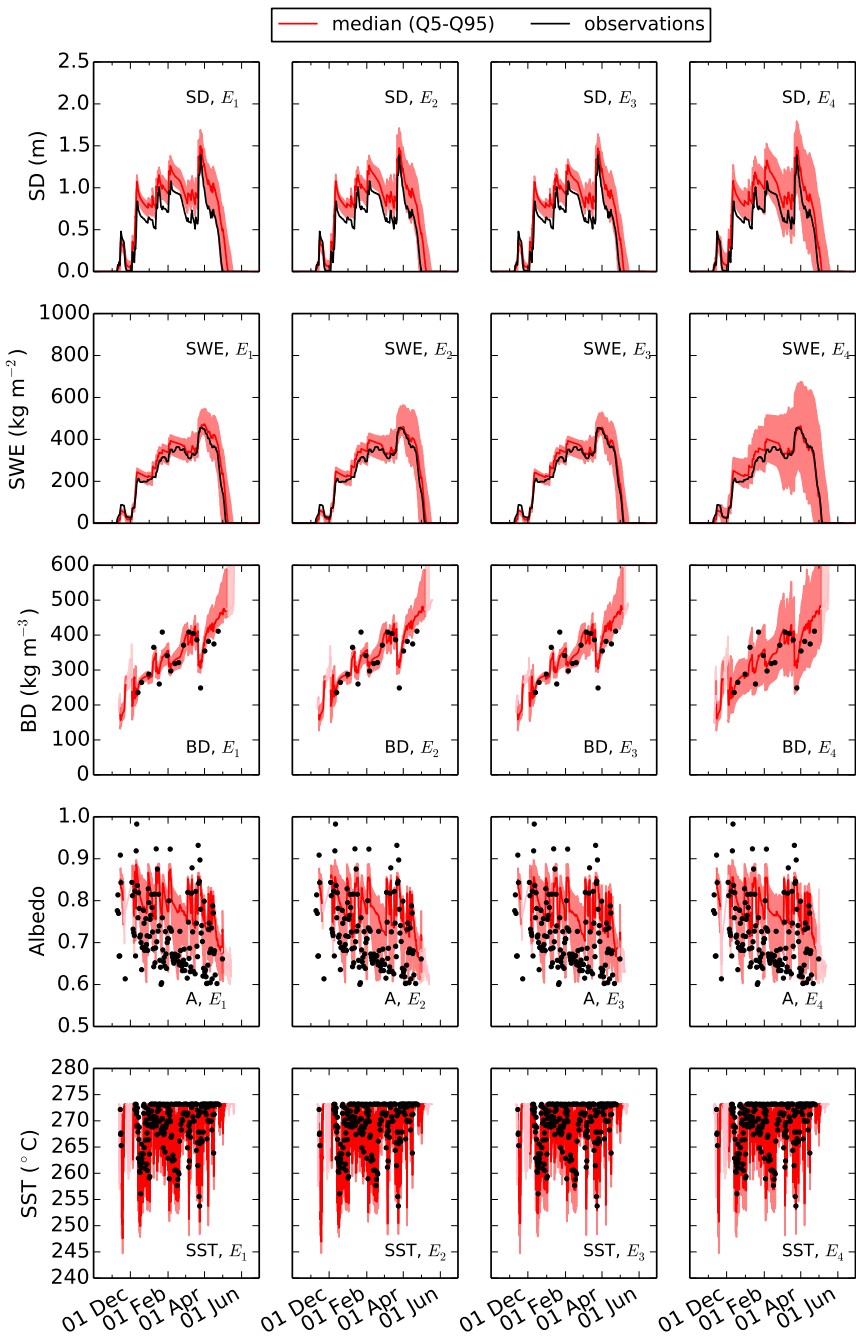

**Figure 10.** Simulated time series (median, 5th and 95th percentiles) of the different evaluation variables (by line), for the different sub-ensembles ($E_1$, $E_2$, $E_3$, $E_4$, by column), season 2007-2008. Black line or black points: observations.



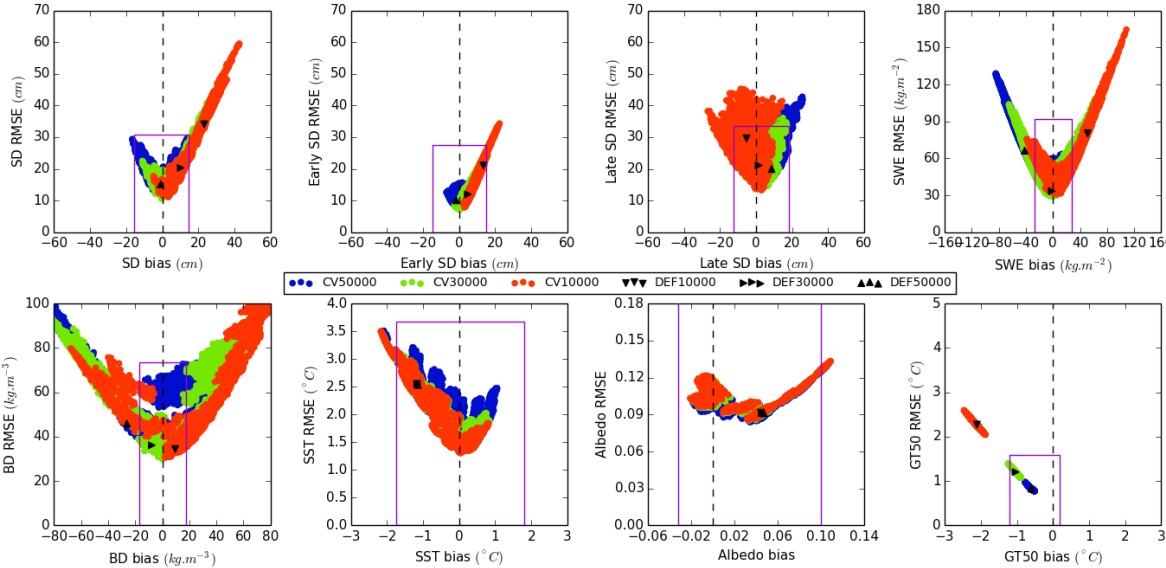

**Figure 11.** Same as Fig. 7 with different colors according to the option of surface heat capacity. The black triangles correspond to the default options of snow processes (blue cells in Fig. 2) and the three options of surface heat capacity.

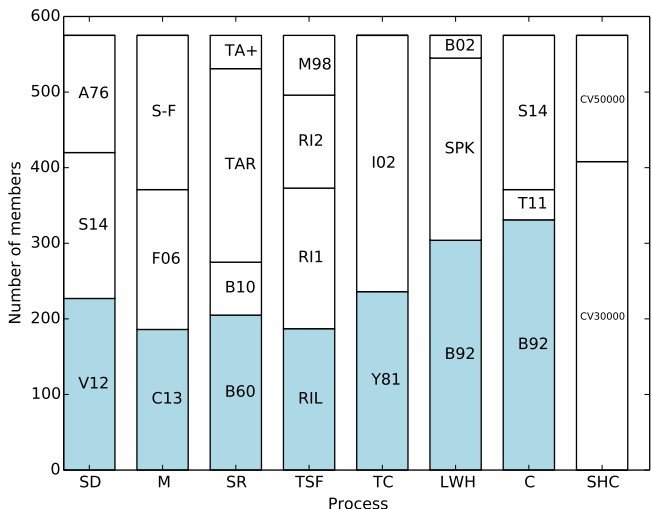

**Figure 12.** Number of occurrences of each physical option in the $E_1$ ensemble of 575 equiprobable members. Blue cells correspond to the default Crocus configuration.

## 5.3 Model sensitivity to physical options




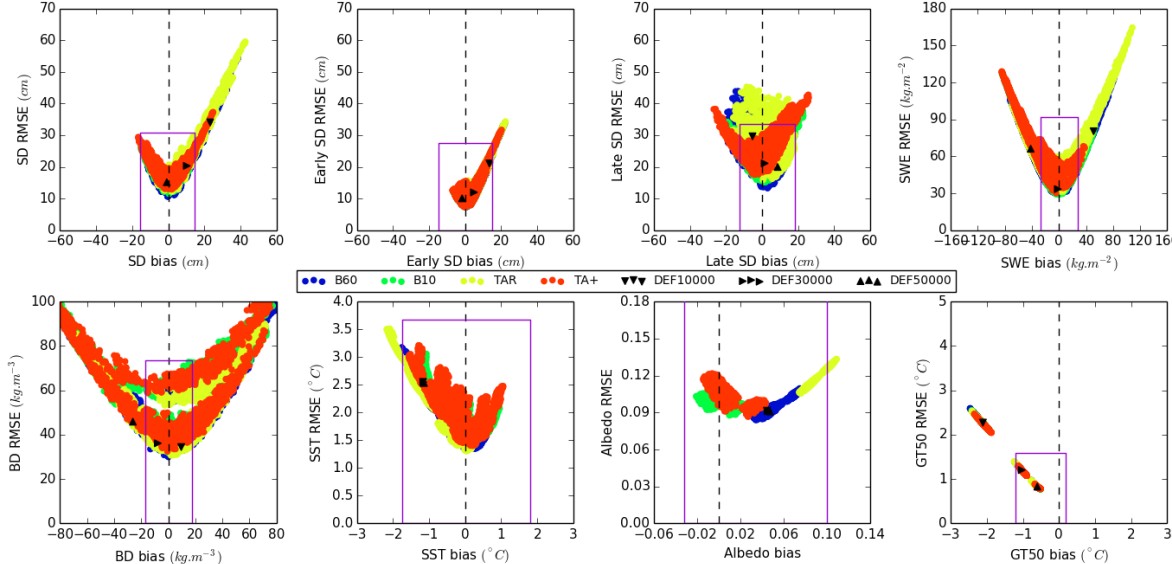

**Figure 13.** Same as Fig. 7 with different colors according to the option of solar radiation absorption. As in Fig. 11, the black triangles correspond to the default options of snow processes (blue cells in Fig. 2) and the three options of surface heat capacity.

It is useful to know if some physical options exhibit a significantly lower skill regardless of the options of the other processes. This is only the case for the default option of $10000\ \mathrm{JK^{-1}kg^{-1}}$ for the soil surface heat capacity which leads to a systematic cold bias of ground temperature in autumn as illustrated in Fig. 11, and in Decharme et al. (2016). As this option exhibits also a high sensitivity on integrated variables such as SD, SWE and BD, further investigations are necessary to better understand

why this realistic value of surface heat capacity leads to a systematic bias at Col de Porte and if this issue also exists in other places in the Alps. For all the other options, their ability to provide optimal simulations depend on the choice of options for the other processes. Fig. 12 represents the frequency of each physical option in the sub-ensemble $E_1$ of 575 optimal members. Regardless of the process, there is no option which is highly prevailing in this sub-ensemble population. Some options are less frequently represented than the others (for instance options TA+ for solar radiation absorption, B02 for liquid water holding,

T11 for snow compaction) but when combined with specific options for the other processes, they are still able to exhibit realistic simulations for all evaluation variables. Regarding the solar radiation absorption representation, it must be noticed that the TA+ and B10 do not exhibit any negative bias in albedo or snow depth evaluations (Fig. 13), their lower frequency in ensemble $E_1$ is mainly explained by the scores of SWE and late snow depth. The B60 option (blue points) could be associated with a positive bias of snow depth in the default Crocus version with a $10000\ \mathrm{JK^{-1}kg^{-1}}$ surface heat capacity (down-facing triangle), but this

bias disappears if the surface heat capacity option is changed. Numerous similar dependences of the skill of a given option to the choice of other processes were found, with interactions between water holding capacity, compaction, and snowfall density options (not shown) or between turbulent surface fluxes and solar radiation absorption options, among others.





To understand how the different options create dispersion in the optimal sub-ensembles, we compare in Fig. 14 for one particular season (2010-2011) the energy fluxes of 2 of the most different members of the $E_2$ optimal sub-ensemble (members 16 and 17 which have different options for all processes except surface heat capacity, see Table A1). The absorbed solar radiation is significantly higher in member 16 with the TA+ option than in member 17 with the B60 option, especially in

February and March, whereas the turbulent heat fluxes are significantly lower in member 16 with RIL option than in member 17 with M98 option. As a result, the temporal variations of the energy balance differs between the two members, with a higher positive balance for member 17 during the windy and mild events of December and January and conversely a higher positive balance during the spring sunny days for member 16. Therefore, there is a different chronology of melting in these members although the final melt-out date difference is only 2 days. This illustrates that very different contributions of energy fluxes

to the energy equilibrium can result in a similar and optimal skill for all evaluated variables (both members are included in $E_2$ sub-ensemble). This equifinality also exists between the other physical processes and options. It explains (i) the difficulty to select a single-model and (ii) the dispersion obtained at a given point in time by several members seen as equivalent and optimal from a deterministic statistical evaluation.

## 6 Discussion

The ability of ESCROC to explain a significant part of modelling errors at Col de Porte and its improved skill in terms of CRPS relatively to deterministic modelling is very promising to contribute to the development of a full ensemble numerical system including ensemble meteorological forcing and ensemble data assimilation techniques in support of avalanche hazard forecasting. However, this work also has several limitations discussed in this section.

### 6.1 Implications of forcing data uncertainty

The assumption that meteorological input errors are low compared to model errors is the main limitation of our study because even at well instrumented sites, recent studies illustrate that model outputs can be significantly affected by input errors especially by long-term biases (Raleigh et al., 2015) and by errors in longwave incident radiations and precipitation amount (Sauter and Obleitner, 2015). The quantification of the impact of meteorological uncertainties in snow modelling by Raleigh et al. (2015) is highly dependent on the choice of the error types (bias vs random errors), and of the error distributions (form

and parameters) whereas the accuracy specified by the manufacturers is difficult to interpret. Furthermore, additional errors can be encountered due to environment specific issues. As a consequence, it is difficult to quantify an absolute contribution of forcing errors in the total uncertainty from manufacturers informations, and new investigations to document the meterological uncertainty in experimental sites must be supported, as recently done for precipitation amounts during the WMO Solid Precipitation Intercomparison Experiment (SPICE, e.g. Kochendorfer et al., 2016) including the Col de Porte site. Nevethe-

less, Raleigh et al. (2015) calculated that the contribution of forcing errors specified from manufacturers may be as high as 40% of the total uncertainty for the most sensitive variables. This suggests that the underdispersion of the sub-ensembles $E_1$, $E_2$ and $E_3$ of equiprobable members might be partly or totally explained by forcing uncertainty and not by an unsufficient





**Figure 14.** Daily simulated energy fluxes (shortwave net radiation SW, longwave net radiation LW, sensible heat flux H and latent heat flux LE) and total energy balance of members 16 and 17 of ensemble $E_2$, and simulated total snow depth of both members. Season 2010-2011.

coverage of the uncertainty of physical processes in ESCROC. This suggests that including lower skill members to optimize the dispersion could be an artificial compensation of not accounting for forcing errors in the simulations. Therefore, despite





the slight enhancement of probabilistic scores, we recommend considering this method with caution. Furthermore, loosing the equiprobability of members can be a disadvantage in practical applications.

## 6.2 Implications of a single site application only

The second limitation of this study is that all evaluations and selections of sub-ensembles were based on a single experimental
site. Larger scales applications of the ESCROC system will be affected by much more significant errors in the meteorological forcing, preventing a large scale calibration of multiphysics sub-ensembles. However, applications of the model in different environments, for instance at higher elevations or higher latitudes, might be affected by different error levels associated to the physical processes than at Col de Porte, and the spread of the proposed sub-ensembles might be insufficient to quantify this higher uncertainty. This is especially the case over areas affected by significant snowdrift. To reduce this issue, a next
important step will be to evaluate ESCROC in contrasted environments using other well-instrumented sites around the world often used for snow models evaluation, e.g. Sodankylä, Finland (Essery et al., 2016) or Weissfluhjoch, Switzerland (WSL, 2015). Forcing and evaluation data gathered by the ESM-SnowMIP initiative (http://www.climate-cryosphere.org/activities/targeted/esm-snowmip) will be used for this purpose.

## 6.3 Limitations of scores and selection methods

Our results illustrate that a realistic number of members can be sufficient to explain the uncertainty range. Nevertheless, the selection of optimal members presented here is affected by an uncertain characterization of evaluation data errors (Sect. 4). The selection is not unique as only a limited number of possible combinations were randomly tested and it can be objective-dependent. This paper presents a framework which may be adapted for a specific application. For instance, ensemble assimilation of remotely-sensed spectral visible reflectances as in Charrois et al. (2016) would probably require to select a sub-ensemble
by albedo spread-skill optimization and as the uncertainty of albedo measurements is high, it would probably benefit from a better modelling of observations errors in both deterministic and probabilistic scores. In a more general context of a full ensemble system of snowpack modelling with various applications, the selection of members might be improved by defining multi-objective probabilistic criteria combining several evaluation variables, or even several evaluation sites. Recent investigations on that topic for the purpose of ensemble meteorological forecasting proposed generalizations of the classical univariate
probabilistic tools (Gneiting et al., 2008; Scheuerer and Hamill, 2015; Thorarinsdottir et al., 2016) that could be tested in ensemble snow modelling. Increasing the number of dimensions in the evaluation framework by incorporating new sites or new variables such as vertical profiles of temperature, density or microstructure properties, is also likely to result in an empty ensemble in the first selection of members with an optimal deterministic skill over all these variables. This would require to also define multivariate deterministic evaluation criteria as in Essery et al. (2013).



## 6.4 Lessons for numerical snowpack modelling

Our results also illustrate the fact that all evaluations of numerical snow modelling are highly impacted by the choice of evaluation variables, the uncertainty of evaluation data and the physical options chosen for all physical processes even if they are beyond the scope of a particular study. As a result, a sensitivity test to choose the best physical representation of a given

physical process can be misleading if the sensitivity to other processes is not explored or if the chosen evaluated variables do not constrain the model sufficiently. These conclusions are fully in accordance with Essery et al. (2013) and should be considered for the evaluation of any parameterization of a physical-based snow model. It was unfortunately rarely done in past evaluations of most snow models.

ESCROC could be seen as a potential tool to quantify the relative contribution of each physical process to the overall uncer-

tainty in snowpack modelling, e.g. using variance analysis methods as in Sauter and Obleitner (2015). This would provide a useful characterization of the system itself but without any guarantee that these contributions are fully representative of snowpack modelling in a general context. Indeed, the results would be totally dependent on the choices made to define the different options. The latter may not accurately quantify the uncertainty process by process and it would be especially challenging to objectively verify the dispersion for each process.

Our results also illustrate the fact that looking for a perfect deterministic snowpack model might never be sufficient and that finding ways to deal with model errors may be as promising as improving the physics description. In the Crocus snowpack model, recent attempts to introduce more sophisticated physics (e.g. metamorphism by Carmagnola et al. (2014) or implementation of the TARTES optical scheme) did not necessary lead to a significant overall skill improvement, as shown here. However, efforts to improve the physics is also in some cases expected to improve the details of the simulated snowpack ver-

tical structure. For instance, solving Richards equations for the liquid water percolation would allow simulating the ponding of liquid water on capillary barriers and crusts with expected positive impacts on the detailed snow stratigraphy (Wever et al., 2015). Including water vapor transfers in snow modelling might also correct the unrealistic density profiles usually obtained in polar regions where this process is significant (Domine et al., 2013). Nevertheless, the modelling of these complex processes are bound to introduce new uncertainties. Therefore, progresses in snow physics understanding and in ensemble modelling

techniques have to be complementary to generally improve the reliability and the usefullness of snow simulations in various applications (e.g. avalanche hazard forecasting, hydrology, glaciers mass balance, climate change impact studies).

## 7   Conclusions

Numerical snow modelling is affected by various sources of errors which reduce its current reliability and utility in operational applications. Following the progresses achieved in meteorology and hydrology, we suggest that three directions in snowpack

modelling should be explored at the same time:

  – improving the quality of the meteorological forcing and the modelling of the physical processes;

  – predicting the uncertainties of the simulations using ensemble frameworks covering all the errors sources;



- assimilating remotely-sensed or conventional snowpack observations to reduce the errors.

While the first point has been the central topic of investigations in the last 30 years, the other two are only emergent. They are also highly linked as the concept of data assimilation relies on an accurate quantification of modelling errors. In this paper, we contribute to both these innovative topics by proposing a new multiphysics ensemble system of snowpack modelling, ESCROC, based on the implementation in the SURFEX/ISBA/Crocus snowpack model of various options for 8 key processes in the evolution of mid-latitude snowpacks. This system was evaluated with both deterministic and probabilistic tools at the Col de Porte well-instrumented site to reduce meteorological forcing errors as much as possible.

The deterministic skill of the ESCROC members cover a wider range than the uncertainty range of evaluation data for snow depth, snow water equivalent, bulk density and ground temperature. It is not the case for albedo and snow surface temperature whose measurements feature higher uncertainty. The results confirm several conclusions of Essery et al. (2013), including the high equifinality between various processes in snow modelling which cannot be eliminated with usual evaluations and should therefore be carefully considered in future works intending to improve snow physics.

To define an ensemble of equiprobable members suited for ensemble forecasting and ensemble assimilation applications, a sub-ensemble of best members for all evaluation variables was selected and a methodological framework to select a reduced number of members was presented. Thus, sub-ensembles of only 35 members are able to explain between 50% and 70% of the total simulation errors. They have a significantly better predictive power than the classical deterministic approach. It is possible to optimize the ensemble spread-skill by including members with a lower deterministic skill, but we do not recommend it because it could be an artificial compensation of not accounting for forcing errors and because it would make the practical use of the ensemble in operational applications more complicated. Several perspectives are listed for future works including the extension of evaluations to a multi-sites and multivariate analysis. Another important perspective will be to test the combination of this multiphysics system with ensemble meterological forecasts suited to snowpack simulations (Vernay et al., 2015). The development of ensemble meteorological analyses instead of current deterministic systems (Durand et al., 1993) will also be a challenge, especially in the context of the increased resolution of NWP models (Vionnet et al., 2016). Last but not least, the development of synthesis diagnostics of ensemble simulations is essential to assist the avalanche hazard forecasters to take advantage of an increasing amount of available data.

## 8   Code availability

ESCROC is developed inside the opensource SURFEX project (http://www.umr-cnrm.fr/surfex). While it is not implemented in an official SURFEX release, the code can be downloaded from the specific branch of the svn repository maintained by Centre d'Études de la Neige. The full procedure and documentation can be found at https://opensource.cnrm-game-meteo.fr/projects/snowtools/wiki/Procedure_for_new_users. For reproductibility of results, the version used in this work is tagged as http://svn.cnrm-game-meteo.fr/projets/surfex/tags/ESCROC-1.0.

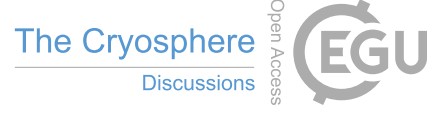

# 9 Data availability

The CDP dataset is placed on the PANGAEA repository (doi:10.1594/PANGAEA.774249) as well as on the public ftp server ftp://ftp-cnrm.meteo.fr/pub-cencdp.

## Appendix A: List of members of sub-ensembles $E_2$, $E_3$ and $E_4$



**Table A1.** List of equiprobable members of sub-ensemble $E_2$ (optimized in terms of snow depth dispersion)

|    | SD  | M   | SR  | TSF | TC  | LWH | C   | SHC     |
|----|-----|-----|-----|-----|-----|-----|-----|---------|
| 1  | V12 | C13 | B60 | RI1 | Y81 | SPK | B92 | CV30000 |
| 2  | V12 | C13 | B60 | RI1 | I02 | B92 | S14 | CV30000 |
| 3  | V12 | C13 | B60 | RI2 | Y81 | B92 | S14 | CV30000 |
| 4  | V12 | C13 | B10 | RIL | Y81 | B92 | B92 | CV30000 |
| 5  | V12 | C13 | TAR | RI1 | I02 | SPK | T11 | CV50000 |
| 6  | V12 | C13 | TAR | RI2 | I02 | SPK | S14 | CV50000 |
| 7  | V12 | C13 | TA+ | RI1 | I02 | SPK | B92 | CV30000 |
| 8  | V12 | F06 | B60 | RIL | Y81 | B92 | S14 | CV30000 |
| 9  | V12 | F06 | B60 | RIL | Y81 | SPK | S14 | CV50000 |
| 10 | V12 | F06 | TAR | RIL | I02 | B92 | T11 | CV50000 |
| 11 | V12 | S-F | B60 | RI1 | I02 | B92 | S14 | CV30000 |
| 12 | V12 | S-F | B10 | RI2 | I02 | SPK | B92 | CV30000 |
| 13 | V12 | S-F | TAR | RIL | Y81 | SPK | S14 | CV50000 |
| 14 | V12 | S-F | TAR | RIL | Y81 | SPK | S14 | CV30000 |
| 15 | V12 | S-F | TAR | M98 | Y81 | SPK | B92 | CV30000 |
| 16 | V12 | S-F | TA+ | RIL | I02 | SPK | B92 | CV30000 |
| 17 | S14 | C13 | B60 | M98 | Y81 | B92 | S14 | CV30000 |
| 18 | S14 | C13 | TA+ | RI1 | I02 | B92 | B92 | CV30000 |
| 19 | S14 | F06 | B60 | RIL | Y81 | B92 | S14 | CV50000 |
| 20 | S14 | F06 | B60 | M98 | I02 | B92 | B92 | CV30000 |
| 21 | S14 | F06 | B60 | M98 | I02 | SPK | B92 | CV30000 |
| 22 | S14 | F06 | TAR | RI1 | Y81 | SPK | B92 | CV30000 |
| 23 | S14 | F06 | TA+ | RIL | I02 | SPK | B92 | CV30000 |
| 24 | S14 | F06 | TA+ | RI1 | I02 | SPK | B92 | CV30000 |
| 25 | S14 | S-F | B60 | RIL | I02 | B92 | B92 | CV50000 |
| 26 | S14 | S-F | B60 | RIL | I02 | SPK | S14 | CV50000 |
| 27 | S14 | S-F | B60 | RI1 | I02 | SPK | B92 | CV30000 |
| 28 | S14 | S-F | TAR | RIL | I02 | SPK | S14 | CV50000 |
| 29 | A76 | F06 | B60 | M98 | I02 | B02 | S14 | CV30000 |
| 30 | A76 | F06 | TAR | M98 | I02 | SPK | B92 | CV30000 |
| 31 | A76 | F06 | TA+ | RIL | I02 | B92 | B92 | CV30000 |
| 32 | A76 | S-F | B10 | RIL | Y81 | B92 | B92 | CV30000 |
| 33 | A76 | S-F | B10 | RI2 | I02 | B92 | B92 | CV30000 |
| 34 | A76 | S-F | TAR | RIL | Y81 | SPK | S14 | CV50000 |
| 35 | A76 | S-F | TAR | RI1 | Y81 | SPK | B92 | CV30000 |



**Table A2.** List of equiprobable members of sub-ensemble $E_3$ (optimized in terms of snow depth CRPS)

|    | SD  | M   | SR  | TSF | TC  | LWH | C   | SHC     |
|----|-----|-----|-----|-----|-----|-----|-----|---------|
| 1  | V12 | C13 | B60 | RIL | I02 | B92 | T11 | CV50000 |
| 2  | V12 | C13 | B60 | RI2 | Y81 | B92 | B92 | CV30000 |
| 3  | V12 | C13 | B60 | RI2 | I02 | SPK | B92 | CV30000 |
| 4  | V12 | C13 | B60 | M98 | I02 | SPK | B92 | CV30000 |
| 5  | V12 | C13 | B10 | RIL | Y81 | SPK | B92 | CV30000 |
| 6  | V12 | C13 | B10 | RI1 | I02 | B92 | B92 | CV30000 |
| 7  | V12 | C13 | TAR | RI2 | Y81 | B92 | S14 | CV30000 |
| 8  | V12 | F06 | B60 | RI2 | Y81 | SPK | B92 | CV30000 |
| 9  | V12 | F06 | TAR | RI1 | I02 | SPK | T11 | CV50000 |
| 10 | V12 | S-F | B60 | RI1 | I02 | SPK | B92 | CV50000 |
| 11 | V12 | S-F | B10 | RIL | Y81 | B92 | B92 | CV30000 |
| 12 | V12 | S-F | TAR | M98 | Y81 | SPK | S14 | CV30000 |
| 13 | V12 | S-F | TA+ | RI1 | I02 | SPK | B92 | CV30000 |
| 14 | S14 | C13 | B60 | RIL | I02 | SPK | B92 | CV50000 |
| 15 | S14 | C13 | TAR | RI2 | I02 | SPK | S14 | CV50000 |
| 16 | S14 | C13 | TAR | M98 | Y81 | B92 | S14 | CV30000 |
| 17 | S14 | F06 | B60 | M98 | Y81 | B92 | S14 | CV30000 |
| 18 | S14 | F06 | TAR | RIL | Y81 | SPK | S14 | CV50000 |
| 19 | S14 | F06 | TA+ | RI1 | I02 | SPK | B92 | CV30000 |
| 20 | S14 | S-F | B60 | RIL | Y81 | B92 | S14 | CV30000 |
| 21 | S14 | S-F | B60 | RI1 | I02 | B92 | S14 | CV50000 |
| 22 | S14 | S-F | B60 | M98 | I02 | B92 | B92 | CV30000 |
| 23 | S14 | S-F | B60 | M98 | I02 | B92 | S14 | CV30000 |
| 24 | S14 | S-F | TAR | RI2 | I02 | B92 | B92 | CV30000 |
| 25 | S14 | S-F | TAR | M98 | I02 | SPK | B92 | CV30000 |
| 26 | S14 | S-F | TA+ | RI1 | I02 | SPK | B92 | CV30000 |
| 27 | A76 | C13 | B60 | RI2 | I02 | B92 | B92 | CV30000 |
| 28 | A76 | C13 | B60 | M98 | I02 | B02 | S14 | CV30000 |
| 29 | A76 | C13 | B10 | RI2 | I02 | B92 | B92 | CV30000 |
| 30 | A76 | C13 | TAR | RI1 | Y81 | B92 | S14 | CV50000 |
| 31 | A76 | F06 | B60 | RI1 | Y81 | B92 | B92 | CV30000 |
| 32 | A76 | F06 | B60 | M98 | Y81 | B02 | S14 | CV30000 |
| 33 | A76 | F06 | B10 | RIL | Y81 | B92 | B92 | CV30000 |
| 34 | A76 | F06 | TAR | RI2 | I02 | B92 | B92 | CV30000 |
| 35 | A76 | S-F | TA+ | RI1 | I02 | B92 | B92 | CV30000 |





**Table A3.** List of non-equiprobable members of sub-ensemble $E_4$ (optimized in terms of snow depth dispersion)

|    | SD  | M   | SR  | TSF | TC  | LWH | C   | SHC     |
|----|-----|-----|-----|-----|-----|-----|-----|---------|
| 1  | V12 | C13 | B60 | RI2 | I02 | SPK | B92 | CV50000 |
| 2  | V12 | C13 | B10 | M98 | Y81 | B02 | B92 | CV30000 |
| 3  | V12 | C13 | TAR | RIL | Y81 | B92 | B92 | CV50000 |
| 4  | V12 | C13 | TA+ | RI1 | Y81 | B92 | B92 | CV50000 |
| 5  | V12 | C13 | TA+ | RI2 | I02 | SPK | T11 | CV50000 |
| 6  | V12 | F06 | B60 | RIL | I02 | SPK | T11 | CV10000 |
| 7  | V12 | F06 | B60 | RI2 | I02 | B92 | S14 | CV50000 |
| 8  | V12 | F06 | TAR | RIL | I02 | SPK | S14 | CV50000 |
| 9  | V12 | S-F | B60 | RI2 | Y81 | B92 | S14 | CV50000 |
| 10 | V12 | S-F | B60 | RI2 | I02 | B02 | S14 | CV50000 |
| 11 | V12 | S-F | B10 | RI1 | I02 | SPK | B92 | CV30000 |
| 12 | V12 | S-F | B10 | M98 | I02 | B02 | S14 | CV10000 |
| 13 | V12 | S-F | TAR | M98 | Y81 | B92 | B92 | CV10000 |
| 14 | S14 | C13 | B60 | RIL | I02 | SPK | S14 | CV30000 |
| 15 | S14 | C13 | B10 | RIL | Y81 | B92 | B92 | CV10000 |
| 16 | S14 | C13 | TAR | RI1 | I02 | B92 | B92 | CV30000 |
| 17 | S14 | F06 | B60 | RI2 | I02 | SPK | S14 | CV50000 |
| 18 | S14 | F06 | B10 | RIL | I02 | B02 | S14 | CV30000 |
| 19 | S14 | F06 | B10 | RI1 | I02 | B92 | B92 | CV30000 |
| 20 | S14 | F06 | TAR | RI2 | I02 | SPK | B92 | CV30000 |
| 21 | S14 | S-F | B60 | RI2 | Y81 | B92 | T11 | CV30000 |
| 22 | S14 | S-F | B60 | RI2 | I02 | SPK | S14 | CV30000 |
| 23 | S14 | S-F | B10 | M98 | Y81 | B92 | B92 | CV10000 |
| 24 | S14 | S-F | TAR | RI2 | I02 | B02 | B92 | CV30000 |
| 25 | S14 | S-F | TAR | M98 | Y81 | B92 | S14 | CV50000 |
| 26 | A76 | C13 | TAR | RI2 | Y81 | SPK | S14 | CV10000 |
| 27 | A76 | C13 | TAR | RI2 | Y81 | B02 | S14 | CV50000 |
| 28 | A76 | C13 | TA+ | RIL | I02 | SPK | T11 | CV10000 |
| 29 | A76 | F06 | B60 | M98 | Y81 | B02 | S14 | CV10000 |
| 30 | A76 | F06 | TAR | RI2 | I02 | B92 | S14 | CV30000 |
| 31 | A76 | F06 | TA+ | M98 | I02 | B92 | B92 | CV30000 |
| 32 | A76 | S-F | B60 | RI1 | Y81 | SPK | B92 | CV10000 |
| 33 | A76 | S-F | B10 | RIL | Y81 | B92 | B92 | CV30000 |
| 34 | A76 | S-F | B10 | M98 | I02 | B02 | B92 | CV30000 |
| 35 | A76 | S-F | TA+ | RI2 | I02 | SPK | B92 | CV10000 |





*Competing interests.* The authors declare that they have no conflict of interest.

*Acknowledgements.* The authors would like to thank all the staff of Centre d'Études de la Neige involved in data acquisition at Col de Porte since 1993, especially J.M. Panel, B. Lesaffre, D. Poncet, E. Le Gac, E. Pougatch, Y. Daniélou, P. David[†] and M. Sudul[†] and EDF-DTG which operate automated SWE measurements. CNRM/CEN is part of LabEx OSUG@2020 (ANR10 LABX56). The CDP observatory belongs to OSUG/CENACLAM, the SOERE CRYOBSCLIM, the GCW CryoNet network and INARCH.



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
