# Peer review of "A multiphysical ensemble system of numerical snow modelling."

_The Cryosphere, 2016_

## Referee Comment (RC1) · R. L. H. Essery (Referee) · 8 Feb 2017

I think that this is an important paper. Several recent studies have also used multiphysical snow models, but they have been rather exploratory in nature, e.g. investigating sensitivity to missing or simplistically represented processes. The different approach of this paper in seeking to construct an ensemble of equally plausible models is a necessary step towards being able to use mulitphysical ensembles to characterise model error for data assimilation.

I have some minor questions, correction and suggestions:

p8, Figure 2

Because there is only one option used for snow drift, and that is not to have snow drift,

it doesn't seem worth having a box for it in this figure.

p11, Table 2

Where does the parameter value $l_f$ = 0.05 m come from? Why not just add dry deposition to the surface layer?

p13, equation 8

It is fairly obvious what $P$ is, but I don't think it has been stated anywhere. Same comment about $\rho_w$ and $\rho_i$.

p14

The number of typos identified in previous papers concerning maximum liquid water holding capacity of snow is striking. This paper itself is not immune. Are the options W14 in section 3.7, S14 in Figure 6 and SPK in Figure 2 all the same thing? Plotting equation 11, I don't get the same curve for C98 as on Figure 6; please check.

p16, equation 18

The dimensions of this equation are wrong, according to the units of the variables given in the text. Incidentally, what are the thickness and heat capacity of the first soil layer, and is freezing of soil moisture allowed for?

p23, Figure 8

The first two columns appear to be identical and both are labelled E2.

p27, Figure 13

Is it worth repeating the black triangles for heat capacity options? The conclusions about the dependency of B60 on heat capacity could be drawn from the same triangles on Figure 11, leaving the possibility of comparing options for solar radiation absorption and defaults in Figure 13.

p29, Figure 14

The argument that equifinality results from counteractions of the extreme TA+/B60 absorbed solar radiation options and RIL/M98 turbulent heat flux options is plausible. Do pairs of members differing only in these options exist within E1? The SD plot could include observations.

---

## Referee Comment (RC2) · Anonymous Referee #2 · 11 Feb 2017

Comments on "A multiphysical ensemble system of numerical snow modelling," by Lafaysse et al.

1. This study constructs an ensemble of plausible multilayer snowpack models for applications in snow and avalanche forecasting. The paper employs a sampling approach across different model configurations in detailed comparison with a large set of observations at a long-term well instrumented midlatitude snow site. The paper examines tradeoffs between accuracy with respect to observations and dispersion of the ensemble, and places the results in the context of measurement and forcing uncertainty.

2. While I am not an expert in this area the study made a plausible case that it is a significant advance on previous work. The manuscript was very clear and instructive in its description of the design and method, and in articulating the limitations of the

approach. The study takes advantages of recent advances from previous authors and extends that work in ways that are well articulated on p.3 (end of Section 1).

3. I have only minor comments on this study but had a couple of questions that the authors could consider on the broader context of this work.

4. How do the authors see this approach being used operationally? Is the basic suggestion that around 30-50 ensemble members could be used to sample the range of independent configurations possible within the model, and that these realizations could be propagated along with initial condition uncertainty in the setting of probabilistic ensemble forecasting?

5. A second question concerns the impact of covariance of errors across evaluation variables shown in Fig. 7. Perhaps the paper addressed the issue and I missed it, but it is not clear if the realizations that lie within the purple boxes are generally independent or if models that have low errors in some variables show low errors in others. Would there be some way of effectively combining the metrics shown in Figure 7 to gain multivariate information?

6. I thought it was insightful for the authors to present information on the surface energy budget that was independent of the tuning/evaluation variables, and would have liked to see more discussion of this. Could this provide other ways of evaluating the independence of realizations?

7. I think it should be made clear in the introduction that this paper clears up several typos of previous papers (p.14).

8. Why is variable availability coming into the definition of equations (21)-(22) but not in (19)-(20)? Are you accounting for variable availability being distinctive in individual model realizations versus in observations? If snow is not on the ground in some realizations but it is in observations, what is done?

9. Could the authors present an explicit expression for the RMSE of the ensemble

mean? My assumption is that RMSE(bar-E) is estimated by (20) but this should be stated.

10. Section 4.3 Could you make this text as accessible as the previous material? If I understand, the key points of this subsection are

* You are selecting a subensemble n' for further evaluation.

* n' is chosen to limit the computational burden in operational applications (although it is not clear why the tuning here will need to be repeated frequently, or what the ultimate applications will be in practice).

* The n' members should be suitably independent by some measure. P.19 l.29, "most appropriate" is vague.

* The n' members should have similar skill and thus have similar likelihood of being correct.

It's hard to understand what the four ensemble are, but it became clearer after re-reading a couple of times. It is not clear why only a lower bound on skill needs to be defined, don't you want the n' to fall within a range of skill levels bounded above and below? Also,

* Not clear why E4 is based on SS and not CRPS.

* Not clear if the whole procedure would be very sensitive to the choice of reference system for CRPS.

Typos/minor technical comments:

p.1 L. 9: observation uncertainties -> observational uncertainty

l.24: since -> from the

l.26: for the different -> for different

p.2: l.3: relatively to . . . -> relative to empirical considerations based on stratigraphy,

surface property measurements, and the outputs . . .

l.4: met by the other other organizations operating -> found by other organizations' operational

l.8: "from the errors of their initial states, which are usually based on analyses or forecasts of NWP models."

l.9: initial conditions -> initial condition

l.10: Then -> In addition

l.11: systems, much coarser -> systems, which are much coarser variability involved -> variability, for example, those involved

l.15: assimilation data -> data assimilation (here and elsewhere)

l.18: also the basis for the confidence -> also increase confidence

l.29: ensemble of snow simulations based on 1701 different combinations of [to avoid implying that others have since built ensembles of 1701 snow simulations]

p.7

l. 5: more affected by -> which is more affected by

p.8 l.4: Crocus default -> The Crocus default

l.8: called S14 -> called S14, [insert comma]

p.13 l.9: IO2 -> The IO2

l.13: role on -> role in

l.21 [and p.15 l.7]: pores -> pore [adjective] or pores -> pores' [possessive]

p.16: l.21: parameter -> parameters

p.18: l.3: to propose -> proposing

l.29: large scales -> large scale

p.21 l.14: fisrt -> first l.28: too -> too many

p.28, l.23 and after:

radiations -> radiation

informations -> information

"including" -> " , which included"

unsufficient -> insufficient

loosing -> losing

Larger scales applications -> Applications on increasingly large scales

associated to -> associated with

require to select -> require selecting

"optimization and as" -> "optimization. Because "

This would require to also define -> This would also require defining

progresses -> progress [twice]

usefullness -> usefulness

equifinality -> equivalence[?]

usual evaluations -> standard evaluation methods

future works -> future work

---

## Author Comment (AC1) · 28 Mar 2017

**Response to interactive comment from Referee #1 (Richard Essery) :**

Authors responses are shown in blue. Proposed changes in the manuscript are reported in bold.

I think that this is an important paper. Several recent studies have also used multiphysical snow models, but they have been rather exploratory in nature, e.g. investigating sensitivity to missing or simplistically represented processes. The different approach of this paper in seeking to construct an ensemble of equally plausible models is a necessary step towards being able to use mulitphysical ensembles to characterise model error for data assimilation.

On behalf of all authors, we thank Richard Essery for the value he found in our work as well as for his detailed and relevant suggestions.

I have some minor questions, correction and suggestions:
p8, Figure 2

Because there is only one option used for snow drift, and that is not to have snow drift, it doesn't seem worth having a box for it in this figure.

We agree and removed this box in Figure 2.

p11, Table 2
Where does the parameter value l f = 0.05 m come from? Why not just add dry deposition to the surface layer?

The e-folding depth parameter $l_f$ comes from the fact that impurities are deposited preferentially at the surface but some may also be deposited below the snow surface (a few cm) because of air circulation and adsorption of impurities on the snow microstructure. Because the thickness of snow layers vary in time, rather than specifying a fixed deposition rate for a given number of upper layers, we assign a characteristic length for the penetration of impurities at and below the snow surface.

This parameter was first introduced in Charrois et al, 2016 and set to 0.05 m. Although rather arbitrary, it was not modified in this paper. From the literature (Clifton et al, 2008) it might be that values around few mm are more physically consistent (characteristic scale of wind pumping effect) However, as illustrated in Fig C1 below, snow depth or snow albedo simulations are almost not sensitive to the e-folding value within [1 mm – 10 cm] range.

[Figure]

*Fig. C1 Simulated snow depth and albedo for 2 simulations based on 10 cm and 1 mm e-folding depths. The simulations are so close that the lines are overlaid most of the time. The differences (black line for albedo and orange line for snow depth) always stay close to 0.*

We modified the manuscript as follow:

*« This formulation and its parameters are rather uncertain as it has not been specifically evaluated against observations. While **the simulations are weakly sensitive to the e-folding depth $l_f$ , the simulated albedo highly depends on the velocity of impurities deposition.** The typical magnitude of the parameters for black carbon [...] »*

p13, equation 8
It is fairly obvious what P is, but I don't think it has been stated anywhere. Same comment about ρ w and ρ i .

We apologize for not having defined these 3 variables. This is modified as follow in the revised manuscript:
Page 12, line 11: *« $\rho_w$ **is the liquid water density (kg m$^{-3}$).** »*
Page 13, line 3: *« **P is the atmospheric pressure (Pa) and** parameter values are given in Table 4. »*
Page 14, line 7: *« where $\varphi = 1 − (\rho − w_{liq})/ \rho_i$ is the snow porosity **and $\rho_i$ the pure ice density (kg m$^{-3}$)** »*

p14
The number of typos identified in previous papers concerning maximum liquid water holding capacity of snow is striking. This paper itself is not immune. Are the options W14 in section 3.7, S14 in Figure 6 and SPK in Figure 2 all the same thing?

Yes, they are. We apologize for this confusion and homogeneized the manuscript with the SPK abbreviation (for SNOWPACK model).

Plotting equation 11, I don't get the same curve for C98 as on Figure 6; please check.

The computation has been checked but we did not find any issue. To plot the liquid water holding capacity as a function of snow density, it is necessary to replace the porosity in equation 11 by equation 10 at saturation. The development gives a second degree polynom. One of the solution gives the liquid water holding capacity as a function of snow density corresponding to the C98 curve in Figure 6.

p16, equation 18
The dimensions of this equation are wrong, according to the units of the variables given in the text.

Thank you for noting this typo. We apologize for the incorrect unit given for the heat capacity which is expressed in $\mathbf{J\ m^{-2}\ K^{-1}}$ in equation 18. The unit was corrected everywhere in sections 3.9 and 5.3. This typo was due to the fact that we commonly assume a mass of 1 kg m$^{-2}$ for low vegetation. In that case, the heat capacity value in J kg$^{-1}$ K$^{-1}$ is equal to the heat capacity expressed in J m$^{-2}$ K$^{-1}$.

Incidentally, what are the thickness and heat capacity of the first soil layer, and is freezing of soil moisture allowed for?

The thickness of the first soil layer is now given in the manuscript (0.01 m). Its heat capacity is computed as the sum of the water heat capacity and the heat capacity of the soil matrix (Decharme et al, 2011). Freezing of soil moisture is allowed (Decharme et al, 2016) but for a better clarity equation 18 is given in the case without any phase change.

Page 16 line 16:
*« where G is the sum of radiative and turbulent energy fluxes at the surface (W m$^{-2}$ ), $\lambda_1$ the thermal conductivity of the first soil layer (W m$^{-1}$ K$^{-1}$ ), $\Delta z_1$ its thickness **(0.01 m)**, $c_{G1}$ its heat capacity (J m$^{-2}$ K$^{-1}$ ) **depending on its water content**, and $T_2$ the temperature of the second soil layer. **Equation 18 corresponds to the case without soil freezing or thawing which are also represented in the model (Decharme et al., 2016).** »*

p23, Figure 8
The first two columns appear to be identical and both are labelled E2.
Thank you again, there was a bug in the preparation of subfigures. This is now corrected. The comments on this figure were not affected by this bug.

p27, Figure 13
Is it worth repeating the black triangles for heat capacity options? The conclusions about the dependency of B60 on heat capacity could be drawn from the same triangles on Figure 11, leaving the possibility of comparing options for solar radiation absorption and defaults in Figure 13.
We modified Fig. 13 to apply this good suggestion and improved in the manuscript the description of the dependency between both processes:

*"The B60 option (blue points) could be associated with a positive bias of snow depth in the default Crocus version with a 10000 J m $-2$ K $-1$ surface heat capacity (down-facing triangle **in Fig. 11 and 13), and the B10 or TA+ options preferred (left and right-facing triangles in Fig. 13). However, an opposite conclusion is obtained if the surface heat capacity option is changed: the positive bias of B60 disappears (right and up-facing triangles in Fig. 11) and a negative late snow depth bias appears in spring for B10 and TA+ (not shown).** Numerous similar dependences of the skill of a given option to the choice of other processes were found [...]"*

p29, Figure 14
The argument that equifinality results from counteractions of the extreme TA+/B60 absorbed solar radiation options and RIL/M98 turbulent heat flux options is plausible. Do pairs of members differing only in these options exist within E1? The SD plot could include observations.

Yes, there are several pairs of members differing only in these options within $E_1$. At first, we thought it might be interesting to illustrate the behaviour obtained by a full set of different physical options. However, it is indeed probably easier to understand equifinality by limiting the differences to 2 processes in this illustration. Therefore, following your remark, we decided to modify this Figure by using 2 members illustrating the equifinality between these 2 processes only. We selected a different year illustrating better this behaviour for these members. Nonetheless, it is important to notice that equifinality can sometimes come from more complex interactions between more than 2 physical options. We added this remark in the revised manuscript. We also added the snow depth observations from the ultra-sound gauge and from the pits on this Figure. This illustrate that the small differences in snow depths between these two members are lower than model errors and than the uncertainty of the reference data. The corresponding paragraph was modified as follows:

*"To illustrate how the different options create dispersion in the optimal sub-ensembles, we compare in Fig. 14 for one particular season (2003-2004) the energy fluxes of **2 different members of the $E_1$ sub-ensemble of optimal members. The two members were selected because they have different options for solar radiation absorption and turbulent fluxes (B60/M98 and TA+/RIL) but the same options for all other processes. The absorbed solar radiation is significantly higher in member with the TA+/RIL options than in member with the B60/M98 options, especially in February and March, whereas the turbulent heat fluxes are significantly lower in member with the TA+/RIL options than in member with the B60/M98 options.** As a result, the temporal variations of the energy balance differs between the two members, with a higher positive balance for member B60/M98 during some windy and mild events in winter and conversely a higher positive balance during some spring sunny days for member TA+/RIL. Therefore, there is a **slightly** different chronology of melting in these members although the final melt-out date difference is only 2 days. **These differences are lower than model errors and lower than the uncertainty range of observations.** This illustrates that very different contributions of energy fluxes to the energy equilibrium can result in a similar and optimal skill for all evaluated variables (both members are included in $E_1$ sub-ensemble). This equifinality also exists between the other physical processes and options, **with some more complex interactions involving more than two processes.** It explains (i) the difficulty to select a single-model and (ii) the dispersion obtained at a given point in time by several members seen as equivalent and optimal from a deterministic statistical evaluation."*

Reference
Clifton, A., Manes, C., Rüedi, JD. et al., 2008, On Shear-Driven Ventilation of Snow, *Boundary-Layer Meteorol* 126: 249. doi:10.1007/s10546-007-9235-0

---

## Author Comment (AC2) · 28 Mar 2017

**Response to interactive comment from Anonymous Referee #2:**

Authors responses are shown in blue. Proposed changes in the manuscript are reported in bold.

1. This study constructs an ensemble of plausible multilayer snowpack models for applications in snow and avalanche forecasting. The paper employs a sampling approach across different model configurations in detailed comparison with a large set of observations at a long-term well instrumented midlatitude snow site. The paper examines tradeoffs between accuracy with respect to observations and dispersion of the ensemble, and places the results in the context of measurement and forcing uncertainty.

2. While I am not an expert in this area the study made a plausible case that it is a significant advance on previous work. The manuscript was very clear and instructive in its description of the design and method, and in articulating the limitations of the approach. The study takes advantages of recent advances from previous authors and extends that work in ways that are well articulated on p.3 (end of Section 1).

On behalf of all authors, we thank Anonymous Reviewer #2 for the value he/she found in our work as well as for his/her detailed and relevant suggestions.

3. I have only minor comments on this study but had a couple of questions that the authors could consider on the broader context of this work.

4. How do the authors see this approach being used operationally? Is the basic suggestion that around 30-50 ensemble members could be used to sample the range of independent configurations possible within the model, and that these realizations could be propagated along with initial condition uncertainty in the setting of probabilistic ensemble forecasting?

The independence of members is not really a necessary condition in ensemble forecasting. The members of ensemble NWP systems based either on stochastic perturbations or on multiphysics are highly correlated (they all share a common model structure and many processes or parameters are not disturbed). As explained in our paper, the important feature is a sufficient dispersion to sample adequately the uncertainty. In this work, we demonstrate that 35 multiphysics members would be sufficient to depict the snowpack model uncertainty at Col de Porte. However, as mentioned in Sect. 6.2, an extension of our evaluations to a large spatial domain is necessary before extrapolating this conclusion over all the French moutain ranges. Furthermore, in a full ensemble system of numerical snow modelling, this uncertainty will have to be combined with the uncertainty of meteorological forcing, for example coming from a meteorological ensemble. In future work, it will be necessary to test if 35 members are sufficient to cover both uncertainties and to study how to combine the meteorological members and the snowpack multiphysics members. (This point is mentioned in the conclusion.) Although beyond the scope of this paper, this is a necessary preliminary step before being able to accurately describe the future ensemble system we plan to build. Note also that as soon as the ensemble system is combined with data assimilation, the most efficient number of members and the way to propagate and reduce uncertainty along the season will also depend on the choice of the data assimilation algorithm. We added the following sentence in the conclusion on that topic:

*« Another important perspective will be to test the combination of this multiphysics system with ensemble meterological forecasts suited to snowpack simulations (Vernay et al., 2015). The development of ensemble meteorological analyses instead of current deterministic systems (Durand et al., 1993) will also be a challenge, especially in the context of the increased resolution of NWP models (Vionnet et al., 2016). **The most appropriate way to propagate and reduce both***

*uncertainties will also have to be investigated when this full ensemble system is combined with an ensemble data assimilation algorithm.* Last but not least, the development of synthesis diagnostics of ensemble simulations is essential to assist the avalanche hazard forecasters to take advantage of an increasing amount of available data. »

5. A second question concerns the impact of covariance of errors across evaluation variables shown in Fig. 7. Perhaps the paper addressed the issue and I missed it, but it is not clear if the realizations that lie within the purple boxes are generally independent or if models that have low errors in some variables show low errors in others. Would there be some way of effectively combining the metrics shown in Figure 7 to gain multivariate information?

There is correlation among the different scores of Fig. 7. For example, the simulated snow depths have a high temporal correlation which is likely to affect the evaluations of early, late and full season snow depth. There is also a direct relationship between snow depth, snow water equivalent, and bulk density. In this work, the evaluation is demanding as the selected members have to be optimal for 8 different evaluation variables. In our case, the correlation between variables is not an issue as we do not combine the different scores in a common metric. However, this possibility is mentioned as a perspective in Sect. 6.3 for both probabilistic and deterministic evaluations. The reviewer is definitely right that these metrics would need to account for the covariance of errors. This is now mentioned in the revised manuscript:

*« In a more general context of a full ensemble system of snowpack modelling with various applications, the selection of members might be improved by defining multi-objective probabilistic criteria combining several evaluation variables, or even several evaluation sites. Recent investigations on that topic for the purpose of ensemble meteorological forecasting proposed generalizations of the classical univariate probabilistic tools (Gneiting et al., 2008; Scheuerer and Hamill, 2015; Thorarinsdottir et al., 2016), which could be tested in ensemble snow modelling. Special care should be taken in the future to deal with the covariance of errors among the different evaluation variables. »*

6. I thought it was insightful for the authors to present information on the surface energy budget that was independent of the tuning/evaluation variables, and would have liked to see more discussion of this. Could this provide other ways of evaluating the independence of realizations?

As previously explained in point 4 of this response letter, independence between members is not a requirement in ensemble modelling. Figure 14 is an example of the uncertainty in the energy budget resulting from the uncertainty of the different physical parameterizations of the Crocus model. Although very different, both energy budgets are still correlated because they are obtained with the same model structure and the same meteorological forcing.

7. I think it should be made clear in the introduction that this paper clears up several typos of previous papers (p.14).

As we implemented several formulations of liquid water capacity, we discovered the typos on the equations in the different papers used as reference. However, the conclusions of these papers are not affected by the minor typos in the equations. Therefore, we consider that the correction of typos is not a key point of our manuscript and we prefer to focus our introduction on explaining the context and goal of our work.

8. Why is variable availability coming into the definition of equations (21)-(22) but not in (19)-(20)? Are you accounting for variable availability being distinctive in individual model realizations

versus in observations? If snow is not on the ground in some realizations but it is in observations, what is done?

The variable availability has to be considered for computing probabilistic scores because at a given instant, the variable can be defined for some members (with snow on the ground) and undefined for some others without snow on the ground anymore. This is not a common issue in ensemble forecasting: in meteorology or hydrology, variables such as precipitation or discharge are always defined for all the members. Therefore, the standard definition of probabilistic scores do not consider incomplete ensembles, and the definition had to be modified in our context.

Conversely, applying deterministic scores on incomplete time series is much more common in various areas. In equations (19) and (20), $N$ corresponds to the number of data used to compute these scores following the variable-dependent restrictions described in the different subsections of Sect. 2.2 and the general restrictions given in the header of this section:

*« To eliminate the summer period without snow on the ground, the time series are limited to the period between October 1st and June 30th . The 0 values of SD and SWE between these two dates are kept for the evaluations to appropriately evaluate the formation and disappearance of the snowpack. BD, A, and SST are not defined when there is no snow on the ground. »*

$N$ is therefore variable between evaluation variables due to the variable availability of observations. $N$ is also variable between members due to the variable length of the snow season. This is not an issue in the definition of the deterministic scores. However, it is true that this is a limitation for the intercomparison of deterministic scores. For a better homogeneity, it would have been possible to remove from the evaluation period any day with 1 member or more without snow on the ground. We did not choose this option because it would have eliminated a very large number of data. However, we decided to better emphasize this limitation in the revised manuscript by indexing $N$ with the $i$ member index ($N_i$) and by adding the following comment:

*« The skill of each singular member is evaluated by deterministic scores comparing $N_i$ simulated values $m_{ki}$ by member $i$ to the corresponding $N_i$ observations $o_k$.* **Note that $N_i$ depends on the observations availability which is specific to each evaluation variable (Sect. 2.2). Furthermore, for variables, which are not defined when there is no snow on the ground (BD, SST, A), $N_i$ is specific to each member $i$ due to the variable duration of the snow season between members.** *We compute the bias estimator $B_i$ and the Root Mean Square Error estimator $RMSE_i$: »*

9. Could the authors present an explicit expression for the RMSE of the ensemble mean? My assumption is that RMSE(bar-E) is estimated by (20) but this should be stated.

We added an equation and slightly reorganized the paragraph:
**« The RMSE of an ensemble corresponds to the RMSE of the ensemble mean $\bar{E}$ over the N dates where observations are available and at least 1 member is defined. »**

$$\mathrm{RMSE}(\bar{E}) = \sqrt{\frac{1}{N} \sum_{k=1}^{N} (\bar{E}_k - o_k)^2}$$

10. Section 4.3 Could you make this text as accessible as the previous material?

Our feeling is that it is difficult to significantly simplify the text without removing important details for the reproductibility of the algorithm and for the justification of our choices. However, we think

that a synthetic table to summarize the different properties of the 4 ensembles will help to improve the understanding of this complex section. We added the following table in the manuscript:

**Table 6.** Summary of the 4 sub-ensembles.

| | $E_1$ | $E_2$ | $E_3$ | $E_4$ |
|---|---|---|---|---|
| Restriction to the best members (determistic scores) | yes | yes | yes | no |
| Restriction to $n' = 35$ members | no | yes | yes | yes |
| Optimization criteria | | SS on SD | CRPS on SD | SS on SD |

If I understand, the key points of this subsection are
* You are selecting a subensemble n' for further evaluation.
Yes.
* n' is chosen to limit the computational burden in operational applications (although it is not clear why the tuning here will need to be repeated frequently, or what the ultimate applications will be in practice).
Yes, n' is chosen to limit the computational burden in operational applications. The selection of the *n'* members will not need to be repeated frequently. But in an operational system, running each day 575 simulation members over large domains is likely to be too much expensive. Furthermore, we demonstrate in our results that it would not be very useful as a similar skill can be obtained with a much lower number of members.
* The n' members should be suitably independent by some measure. P.19 l.29, "most appropriate" is vague.
The n' members do not really need to be independent as previously explained (point 4 of this response letter). They need to exhibit a dispersion close to the magnitude of the root mean square error for all the simulated variables. However, it is likely that the more independent the members, the higher the dispersion. We remove 'most appropriate' in the revised manuscript. The Spread-Skill is a natural candidate to measure this behaviour. However, a very high dispersion with a very high RMSE would give a good Spread-Skill but it would not be a satisfactory ensemble (poor skill of the mean). This is why the optimization of two different metrics is tested for members selection (Spread-Skill or CRPS).
* The n' members should have similar skill and thus have similar likelihood of being correct.
Absolutely.
It's hard to understand what the four ensemble are, but it became clearer after rereading a couple of times.
We hope that the new table will be helpful for a quicker understanding.
It is not clear why only a lower bound on skill needs to be defined, don't you want the n' to fall within a range of skill levels bounded above and below?
We do not see any argument to apply a higher bound of skill. The best members in a determistic point of view are likely to have value in the ensemble and to contribute to reduce the CRPS. This is why only unrealistic members are excluded.

Also,
* Not clear why E4 is based on SS and not CRPS.
$E_4$ is based on SS for comparison with sub-ensemble $E_2$ to test the impact of the initial restriction to optimal members (which is done in $E_2$ but not in $E_4$). We thought that describing a fifth ensemble in the paper, based on CRPS and without the initial restriction to optimal members, would have increased too much its complexity.

* Not clear if the whole procedure would be very sensitive to the choice of reference system for CRPS.

The choice of the reference member only affects the CRPSS computation, not the CRPS. In the members selection procedure for ensemble $E_3$, we optimize the CRPS, so the choice of the reference member has no impact.

Typos/minor technical comments:
p.1 L. 9: observation uncertainties -> observational uncertainty
Corrected.
l.24: since -> from the
Corrected.
l.26: for the different -> for different
We refer to the evelations and slopes already mentioned line 22.
p.2: l.3: relatively to . . . -> relative to empirical considerations based on stratigraphy, surface property measurements, and the outputs . . .
Corrected.
l.4: met by the other other organizations operating -> found by other organizations' operational
« met » corrected by « found ». However, the issues are found by the organizations, not by the systems.
l.8: "from the errors of their initial states, which are usually based on analyses or forecasts of NWP models."
We did not take this suggestion of modification because the analysis systems like SAFRAN do not have an initial state, they have a guess continuous in time which is adjusted by the assimilation of complementary observations. When the guess comes from the forecasts of a NWP model, it is affected by both the uncertain initial conditions of the NWP model and by the physical errors in the NWP model.
l.9: initial conditions -> initial condition
« initial states » was preferred here.
l.10: Then -> In addition
Corrected.
l.11: systems, much coarser -> systems, which are much coarser variability involved -> variability, for example, those involved
Corrected.
l.15: assimilation data -> data assimilation (here and elsewhere)
Corrected.
l.18: also the basis for the confidence -> also increase confidence
Corrected.
l.29: ensemble of snow simulations based on 1701 different combinations of [to avoid implying that others have since built ensembles of 1701 snow simulations]
Corrected.
p.7 l. 5: more affected by -> which is more affected by
Corrected.
p.8 l.4: Crocus default -> The Crocus default
Corrected.
l.8: called S14 -> called S14, [insert comma]
Corrected.
p.13 l.9: IO2 -> The IO2
Corrected.
l.13: role on -> role in
Corrected.
l.21 [and p.15 l.7]: pores -> pore [adjective] or pores -> pores' [possessive]
Corrected. [pores' volume, pores' structure]
p.16: l.21: parameter -> parameters
Corrected.

p.18: l.3: to propose -> proposing
Corrected.
p.19: l.29: large scales -> large scale
Corrected.
p.21 l.14: fisrt -> first
Corrected.
l.28: too -> too many
Corrected.
p.28, l.23 and after:
radiations -> radiation
Corrected.
informations -> information
Corrected.
"including" -> " , which included"
Corrected.
unsufficient -> insufficient
Corrected.
loosing -> losing
Corrected.
Larger scales applications -> Applications on increasingly large scales
Corrected.
associated to -> associated with
Corrected.
require to select -> require selecting
Corrected.
"optimization and as" -> "optimization. Because "
Corrected.
This would require to also define -> This would also require defining
Corrected.
progresses -> progress [twice]
Corrected.
usefullness -> usefulness
Corrected.
equifinality -> equivalence[?]
The term equifinality is commonly used in hydrological modelling (Beven, 2006). It seems appropriate as well in our context.
usual evaluations -> standard evaluation methods
Corrected.
future works -> future work
Corrected.

Reference
Beven, K., 2006. A manifesto for the equifinality thesis. J. Hydrol. 320, 18–36 (3rd MOPEX Workshop, Sapporo, JAPAN, JUL, 2003)